# RankMatch: A Novel Approach to Semi-Supervised Label Distribution Learning Leveraging Rank Correlation between Labels

**Zhiqiang Kou**[1,2], **Yucheng Xie**[1,2], **Hailin Wang**[5], **Junyang Chen**[1,2], **Jing Wang**[1,2],
**Ming-Kun Xie**[3], **Shuo Chen**[3], **Yuheng Jia**[1,2]*, **Tongliang Liu**[4], **Xin Geng**[1,2]*

[1]School of Computer Science and Engineering, Southeast University, China
[2]Key Laboratory of New Generation Artificial Intelligence Technology and Its Interdisciplinary Applications (Southeast University), Ministry of Education, China
[3]RIKEN Center for Advanced Intelligence Project (AIP), Tokyo, Japan
[4]Sydney AI Centre, The University of Sydney, Australia
[5]School of Mathematics and Statistics, Xian Jiaotong University, China

```
{zhiqiang_kou, xieyc, chenjunyang, wangjing91, yhjia, xgeng}@seu.edu.cn,
       {ming-kun.xie, shuo.chen.ya}@riken.jp, wanghailin97@163.com,
                     tongliang.liu@sydney.edu.au
```

## Abstract

Pseudo label based semi-supervised learning (SSL) for single-label and multi-label classification tasks has been extensively studied; however, semi-supervised label distribution learning (SSLDL) remains a largely unexplored area. Existing SSL methods fail in SSLDL because the pseudo-labels they generate only ensure overall similarity to the ground truth but do not preserve the ranking relationships between true labels, as they rely solely on KL divergence as the loss function during training. These skewed pseudo-labels lead the model to learn incorrect semantic relationships, resulting in reduced performance accuracy. To address these issues, we propose a novel SSLDL method called *RankMatch*. *RankMatch* fully considers the ranking relationships between different labels during the training phase with labeled data to generate higher-quality pseudo-labels. Furthermore, our key observation is that a flexible utilization of pseudo-labels can enhance SSLDL performance. Specifically, focusing solely on the ranking relationships between labels while disregarding their margins helps prevent model overfitting. Theoretically, we prove that incorporating ranking correlations enhances SSLDL performance and establish generalization error bounds for *RankMatch*. Finally, extensive real-world experiments validate its effectiveness.

## 1 Introduction

Label Distribution Learning (LDL) [2] [10, 26] is a machine learning paradigm designed to address label ambiguity [11, 53]. Unlike Multi-label Learning [63, 30], which assigns a fixed number of labels to each instance [59], LDL extends this framework by quantifying the importance of each label through description degrees [20], thereby providing richer supervision information [19, 28].

---

*Corresponding authors.

[2]LDL is similar to learning from soft labels, but the soft-label formulation focuses on single-label problems (i.e., there is only one true label for each instance), while LDL considers multi-label problems (i.e., each instance can have multiple true labels).

39th Conference on Neural Information Processing Systems (NeurIPS 2025).

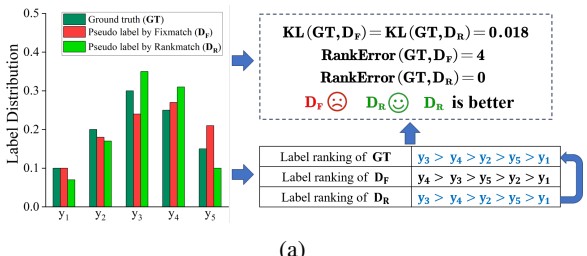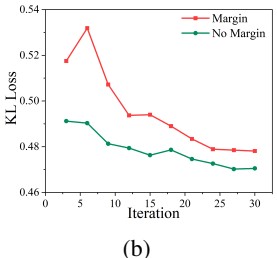

(a)                                        (b)

Figure 1: Illustration of the effectiveness of rank-aware pseudo-labeling. (a) Comparison of Pseudo-Labels by FixMatch ($\mathbf{D}_F$) and RankMatch ($\mathbf{D}_R$) on a sample from the Twitter-LDL dataset [61]. $\mathbf{D}_F$ fails to preserve the true label ranking (**GT**), despite a low KL divergence. In contrast, $\mathbf{D}_R$ maintains ranking relationships with higher Kendall tau ($\tau$). (b) Performance on RAF-LDL dataset [24]. Flexible pseudo-label utilization improves performance by focusing on ranking relationships without strict margin alignment.

Deep learning has demonstrated remarkable success across various domains, primarily due to its ability to leverage large-scale and accurately labeled datasets [59, 39], which are essential for training deep neural networks (DNNs) with strong generalization. However, obtaining labeled data for LDL is particularly challenging and costly [31, 29]. For example, annotating the RAF-LDL dataset [34] required 315 trained annotators, with each image annotated multiple times to generate appropriate label distributions [34, 27]. This highlights the significant burden of creating labeled datasets for LDL. Given these challenges, the importance of semi-supervised LDL becomes evident.

Semi-supervised learning (SSL) [1, 9] has made significant progress, particularly in the areas of single-label [52, 23] and multi-label learning [59, 39] based on classical deep learning. However, Semi-Supervised Label Distribution Learning (SSLDL) remains relatively underexplored. One of the key techniques in SSL is leveraging trained models to generate pseudo-labels [56, 33] for unlabeled data, with the most well-known methods being FixMatch [52] and MixMatch [2]. These methods [52, 23] fail in SSLDL because, during training with labeled data, the model focuses solely on minimizing the overall similarity between predicted and ground truth label distributions (e.g., using KL divergence as loss function for training) without learning label ranking correlations, leading to biased pseudo-labels. As shown in Fig. 1(a), for a sample from the Twitter-LDL dataset [61], FixMatch generates pseudo-labels with a low KL divergence (KL = 0.018) but entirely incorrect label ranking correlations. This phenomenon is common across other datasets: ignoring the ranking relationships among labels during training produces distorted pseudo-label distributions (PLDs), and training with these PLDs causes the models performance to degrade. Moreover, we found that forcing the model to exactly match the pseudo-label distributions during training leads to overfitting. In contrast, using only the relative ranking among pseudo-labels preserves the underlying structure more robustly. Experiments on the RAF-LDL dataset (Fig. 1(b)) demonstrate that imposing only ranking constraints on pseudo-labelsrather than enforcing strict numeric matchingsignificantly improves model performance.

In this paper, we first propose a pseudo-label-based SSLDL method called *RankMatch*, incorporates the ranking correlations between labels into the supervised training process. We introduce a novel loss function called the *Pairwise Ranking Relationship Loss (PRR Loss)* to enhance the ability of pseudo-labels to capture the ranking correlations between labels. Furthermore, we introduce a flexible pseudo-label training strategy that prioritizes ranking relationships between labels while disregarding margins, which prevents the model from overfitting to absolute label differences and enables a more robust utilization of pseudo-labels. In the theoretical aspect, we prove that incorporating ranking correlations between labels can enhance the performance of SSLDL and provide generalization error bounds for the *RankMatch*. Finally, extensive experiments on real-world datasets validate the effectiveness of our method. Our contributions can be summarized as follows:

- To the best of our knowledge, this is the first deep learning-based SSLDL algorithm utilizing pseudo-labels. Compared to existing SSL methods, our approach generates pseudo-labels that better align with the label distribution setting. Additionally, we propose a flexible pseudo-label utilization strategy for SSLDL.

- We theoretically demonstrate that incorporating label ranking correlations enhances model performance and provide a generalization bound for *RankMatch*.

- Extensive experiments on multiple datasets validate the effectiveness of *RankMatch*, consistently outperforming existing *SSLDL* methods.

## 2   RELATED WORK

Label Distribution Learning (LDL) [10, 54] assigns a distribution over labels to each instance, establishing a direct mapping between instances and their label distributions. Originally proposed for facial age estimation [12], LDL generates distributions across all possible age categories, offering richer supervisory signals than traditional single-label approaches. This paradigm has also demonstrated strong performance in facial emotion recognition, where it effectively models ambiguous emotional states by capturing uncertainty within the label space [49, 48, 38, 47].

Beyond facial analysis, LDL has shown broad applicability across diverse domains. For instance, NASA employed LDL to infer the chemical compositions of Martian meteorites [42], refining the method to predict elemental abundances from crystallographic data. In mental health, LDL has been used for depression detection via the Deep Joint Label Distribution and Metric Learning framework, which identifies subtle variations in facial expressions associated with different depression levels [66]. In crowd analysis, Ling [35] applied LDL to estimate indoor crowd densities by assigning label distributions that more accurately describe population levels in video frames.

Despite its success, LDL still faces challenges due to the scarcity of precisely annotated data [37, 60]. To mitigate this, several Semi-Supervised Label Distribution Learning (SSLDL) methods have been proposed. Hou [18] inferred the label distribution of unlabeled data by averaging the labels of its nearest neighbors, using both labeled and unlabeled samples for training. Jia [22] enhanced label distribution recovery by exploiting graph-structured relationships among instances. Liu [39] further developed a co-regularization-based SSLDL framework that leverages dual model structures to improve robustness and consistency.

However, these SSLDL methods are often not end-to-end and rely heavily on manual feature engineering, limiting their scalability to high-dimensional or large-scale data. They also underutilize unlabeled information. In contrast, deep learning provides a natural mechanism for automatic representation learning and has demonstrated remarkable success in data-rich environments. Consequently, integrating deep learning into SSLDL offers a promising direction to address existing limitations and unlock the full potential of label distribution learning under limited supervision.

## 3   Problem Statement and Notation

In SSLDL, the training data consists of a labeled dataset $\mathcal{D}_L = \{(\mathbf{x}_i, \mathbf{d}_i) | i = 1, 2, ..., n\}$ and an unlabeled dataset $\mathcal{D}_U = \{\mathbf{x}_g | g = 1, 2, ..., m\}$. Here, $n$ and $m$ represent the number of labeled and unlabeled samples, respectively. In the labeled dataset $\mathcal{D}_L$, $\mathbf{x}_i$ is a labeled sample, and $\mathbf{d}_i = \{d_{\mathbf{x}_i}^{y_1}, d_{\mathbf{x}_i}^{y_2}, ..., d_{\mathbf{x}_i}^{y_c}\}$ is the corresponding label distribution, where $d_{\mathbf{x}_i}^{y_j}$ represents the importance or relevance of label $y_j$ to sample $\mathbf{x}_i$. The label distribution satisfies the normalization constraint $\sum_{j=1}^{c} d_{\mathbf{x}_i}^{y_j} = 1$. $c$ denotes the number of labels in the label space $\mathcal{Y} = \{y_1, y_2, ..., y_c\}$.

## 4   The Method

### 4.1   The Supervised Training Phase

In LDL, we transition from using the traditional binary cross-entropy loss, commonly employed in multi-label learning [17], to adopting Kullback-Leibler (KL) divergence as the loss function. This transition is essential because LDL predicts continuous real-valued label distributions instead of discrete binary outcomes. The KL divergence [17] is well-suited for measuring the difference between the ground-truth and predicted label distributions. The supervised loss is formulated as:

$$\mathcal{L}_s = \frac{1}{n} \sum_{i=1}^{n} \sum_{j=1}^{c} d_{\mathbf{x}_i}^{y_j} \ln \left( \frac{d_{\mathbf{x}_i}^{y_j}}{h(y_j \mid \text{Aug}_w(\mathbf{x}_i); \theta)} \right), \tag{1}$$

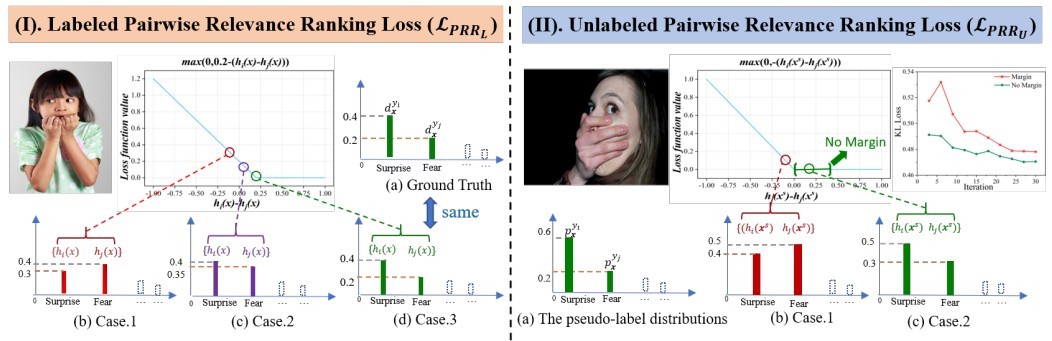

Figure 2: An example to illustrate the $\mathcal{L}_{PRR}$ loss.

where $\text{Aug}_w(\mathbf{x}_i)$ represents a weakly augmented version of the $i$-th labeled sample [52]. The term $h(y_j \mid \text{Aug}_w(\mathbf{x}_i); \theta)$ denotes the predicted importance degree of label $y_j$ for the augmented instance $\text{Aug}_w(\mathbf{x}_i)$, as determined by the model. This is computed as:

$$h(y_j|\mathbf{x}_i; \theta) = \frac{\exp(f_j(\mathbf{x}_i; \theta))}{\sum_{q=1}^{c} \exp(f_q(\mathbf{x}_i; \theta))}, \tag{2}$$

where $f_j(\mathbf{x}_i; \theta)$ represents the raw output of the DNN for label $y_j$ with respect to instance $\mathbf{x}_i$. This formulation ensures that the predicted label distribution $h(y_j \mid \mathbf{x}_i; \theta)$ satisfies the normalization constraint $\sum_{j=1}^{c} h(y_j \mid \mathbf{x}_i; \theta) = 1$.

Existing SSL methods like FixMatch [52] and MixMatch [2] fail in SSLDL because they focus only on minimizing the KL divergence between predicted and ground truth distributions, neglecting label ranking relationships. This oversight leads to pseudo-labels that may have low KL divergence but incorrect label rankings, as shown in the example where FixMatch produces reversed label importance. In contrast, SSLDL requires preserving both the absolute importance and the relative ranking of labels to ensure semantic consistency and accurate predictions.

To produce more reliable pseudo-labels, we propose the Pairwise Relevance Ranking (PRR) loss $\mathcal{L}_{PRR}$, which aligns predictions with the inherent semantic structure of label distributions. For labeled data, $\mathcal{L}_{PRR_L}$ strictly enforces alignment between the predicted and ground-truth label rankings while preserving meaningful margins. For example, when label description degrees $d_{\mathbf{x}}^{y_i} = 0.32$ and $d_{\mathbf{x}}^{y_k} = 0.33$, their negligible difference, likely caused by annotation noise, avoids unnecessary ranking adjustments. Let $h_j(\mathbf{x}_i)$ denote the predicted relevance for the $j$-th label after weak augmentation $\text{Aug}_w$. The $\mathcal{L}_{PRR_L}$ loss is defined as:

$$\mathcal{L}_{PRR_L} = \sum_{1 < j < k < c} \Big( s(j,k) \cdot g_\delta(j,k) + s(k,j) \cdot g_\delta(k,j) \Big), \tag{3}$$

Here, $\delta = d_{\mathbf{x}_i}^{y_j} - d_{\mathbf{x}_i}^{y_k}$, and $f(j,k)$ and $g_\delta(j,k)$ are defined as follows:

$$s(j,k) = \begin{cases} 1, & \text{if } d_{\mathbf{x}_i}^{y_j} > d_{\mathbf{x}_i}^{y_k} \text{ and } d_{\mathbf{x}_i}^{y_j} - d_{\mathbf{x}_i}^{y_k} > t, \\ 0, & \text{otherwise.} \end{cases} \tag{4}$$

$$g_\delta(j,k) = \begin{cases} 0, & \text{if } h_j(\mathbf{x}_i) - h_k(\mathbf{x}_i) \geq \delta, \\ \delta - (h_j(\mathbf{x}_i) - h_k(\mathbf{x}_i)), & \text{otherwise.} \end{cases} \tag{5}$$

Fig. 2(a) shows an example of the $\mathcal{L}_{PRR_L}$ loss, using a sample from the RAF-LDL dataset. This loss penalizes two key scenarios (i) when the predicted ranking of labels deviates from the ground truth (Case 1); (ii) when the ranking is correct but the difference between label scores does not match the ground-truth margin (Case 2). Only in (iii), where both the ranking and the margin exactly match the ground truth (Case 3), does the loss drop to zero, indicating a perfectly correct prediction.

## 4.2 Self-Training Phase by Pseudo-label distribution

*Pseudo-label distribution generation*: To improve prediction stability and effectively utilize unlabeled data, we adopt an ensemble learning-based approach [67]. This method generates pseudo-label distributions (PLDs) for unlabeled instances by averaging the model's outputs from multiple weak augmentations of the same image [52].

The pseudo-label generation process is as follows: given an unlabeled image $\mathbf{x}$, the model computes raw outputs (logits) for $H$ weakly augmented versions $\mathrm{Aug}_w(\mathbf{x})$. The PLD for $\mathbf{x}$, denoted as $\mathbf{p}_i$, is defined as:

$$\mathbf{p}_i(y_j) = \frac{\exp\left(\frac{1}{H}\sum_{k=1}^{H} f_j(\mathrm{Aug}_w(\mathbf{x})_k; \theta)\right)}{\sum_{q=1}^{c} \exp\left(\frac{1}{H}\sum_{k=1}^{H} f_q(\mathrm{Aug}_w(\mathbf{x})_k; \theta)\right)}, \tag{6}$$

where $f_j(\mathrm{Aug}_w(\mathbf{x})_k; \theta)$ is the models raw output (logit) for label $y_j$ on the $k$-th weak augmentation of $\mathbf{x}$.

Then, we define the unsupervised consistency loss, $\mathcal{L}_{uc}$, which aligns the PLD with the predictions on strongly augmented versions of the same instances [52]. It is expressed as:

$$\mathcal{L}_{uc} = \frac{1}{m}\sum_{u=1}^{m}\sum_{j=1}^{c} p_{\mathbf{x}_u}^{y_j} \ln\left(\frac{p_{\mathbf{x}_u}^{y_j}}{h\left(y_j \mid \mathrm{Aug}_s(\mathbf{x}_u); \theta\right)}\right), \tag{7}$$

where $h\left(y_j \mid \mathrm{Aug}_s(\mathbf{x}_u); \theta\right)$ is the predicted importance degree for label $y_j$ after applying strong augmentation to $\mathbf{x}_u$. This loss encourages the model to exploit the underlying structure of the unlabeled data, improving learning from these instances.

In the unsupervised component, we adopt a more flexible strategy to utilize PLDs for training. Recognizing the potential inaccuracies in pseudo-labels, we focus on aligning the ranking relationships among labels rather than enforcing strict adherence to absolute values. To achieve this, we propose the unsupervised pairwise relevance ranking loss, $\mathcal{L}_{PRR_u}$, which prioritizes capturing inter-label ranking while ignoring margins. Let $h_j(\mathbf{x}_i^s)$ denote the predicted relevance of the $j$-th label after strong augmentation $\mathrm{Aug}_s$. The loss is defined as:

$$\mathcal{L}_{PRR_u} = \sum_{1 < j < k < c} \left(s(j,k)\cdot g_0(j,k) + s(k,j)\cdot g_0(k,j)\right), \tag{8}$$

where:

$$s(j,k) = \begin{cases} 1, & \text{if } p_{\mathbf{x}_i}^{y_j} > p_{\mathbf{x}_i}^{y_k} \text{ and } p_{\mathbf{x}_i}^{y_j} - p_{\mathbf{x}_i}^{y_k} > t, \\ 0, & \text{otherwise.} \end{cases} \tag{9}$$

$$g_0(j,k) = \begin{cases} 0, & \text{if } h_j(\mathbf{x}_i^s) - h_k(\mathbf{x}_i^s) \geq 0, \\ h_k(\mathbf{x}_i^s) - h_j(\mathbf{x}_i^s), & \text{otherwise.} \end{cases} \tag{10}$$

As shown in Fig. 2(b), we illustrate the unlabeled Pairwise Relevance Ranking loss $\mathcal{L}_{\mathrm{PRR}_u}$. In this examplea sample from the RAF-LDL datasetthe pseudolabel distribution is $(0.6, 0.2, \dots)$. The loss only penalizes cases where the predicted ranking conflicts with the order suggested by the pseudolabel distribution, i.e., when a label with higher pseudolabel score is ranked lower(Case.1), and it does not impose any margin constraints. Our experiments demonstrate that this flexible use of pseudolabels significantly improves SSLDL performance, as evidenced by the faster convergence and lower KL divergence shown in the topright inset of Fig. 2(b).

**Finally Loss Function**: Overall, the RankMatch algorithm utilizes a dual-phase training strategy to effectively differentiate between labeled and unlabeled data. The combined application of supervised and unsupervised ranking losses under the PRR framework is modulated by a hyperparameter $\lambda$. The total loss is computed as follows:

$$\mathcal{L}_{total} = \mathcal{L}_s + \mathcal{L}_{uc} + \lambda(\mathcal{L}_{PRR_L} + \mathcal{L}_{PRR_u}), \tag{11}$$

## 5 Theoretical Analysis

In this section, we first investigate how the proposed PRR loss influences the generalization behavior of the SSLDL framework. Intuitively, incorporating PRR encourages the model to capture inter-label correlations and refine label ranking consistency, which helps the network generalize beyond

the labeled set. To formalize this intuition, we derive the following theorem, which shows that adding the PRR term leads to a tighter generalization bound compared with the KL-only objective.

**Theorem 5.1.** *Let $\mathcal{F}$ be a hypothesis class of scoring functions, and define the empirical risks $\widehat{R}_{\mathrm{KL}}$, $\widehat{R}_{\mathrm{PRR}}$, and the combined risk $\widehat{R}_{\mathrm{tot}}$ as above. Denote the corresponding minimizers $f_{\mathrm{KL}}$ and $f_{\mathrm{KL+PRR}}$. Then, with probability at least $1 - \delta$,*

$$R_{\mathrm{tot}}(f_{\mathrm{KL+PRR}}) < R_{\mathrm{tot}}(f_{\mathrm{KL}}) + 2\,\mathfrak{R}_{n+m}(\ell \circ \mathcal{F}) + B\sqrt{\frac{\ln(1/\delta)}{2(n+m)}}.$$

This result theoretically confirms thatwhen the PRR term effectively reduces the empirical total risk the overall population risk under PRR regularization becomes strictly smaller than that of the KL-only formulation, up to the standard complexity and confidence terms. In other words, the PRR loss not only improves empirical optimization but also strengthens the generalization guarantee of the model. This provides a theoretical foundation for the performance gains observed in our experiments.

Next we establish a theoretical foundation for our RankMatch by defining a generalization bound.

**Theorem 5.2.** *Let $f^*$ be the true risk minimizer and $\hat{f}$ the empirical risk minimizer. Assume the loss function $\ell(\cdot)$ is bounded by $B$ and that the pseudo-labeling error $\epsilon$ satisfies $\sum_{j=1}^{m} \mid \mathbb{I}(f_k(\mathbf{x}_j)) - \mathbb{I}(d_{\mathbf{x}_j}^{y_k}) \mid /m \leq \epsilon$ for all $k \in [q]$. For a given Lipschitz constant $L_E$, Rademacher complexity $R_N(\mathcal{F})$ of the function class $\mathcal{F}$, and confidence parameter $\delta > 0$, the generalization gap is bounded as:*

$$R(\hat{f}) - R(f^*) \leq 2qB\epsilon + 4qL_E R_N(\mathcal{F}) + 2qB\sqrt{\frac{\log \frac{2}{\delta}}{2N}},$$

where $N = m + n$ is the total number of labeled and unlabeled samples. Theorem 3.2 provides a theoretical guarantee on the performance of the proposed *RankMatch* algorithm. Furthermore, it highlights key factors influencing the generalization error in SSLDL, including the pseudo-labeling error $\epsilon$, the complexity of the hypothesis space captured by the Rademacher complexity $R_N(\mathcal{F})$, and the total number of training samples $N$. Moreover, increasing the training set size $N$ further tightens the bound, reinforcing the benefits of leveraging large-scale unlabeled data in SSLDL. All the proof detail can be find in Appendix A and B.

## 6 Experiments

**Experimental Datasets.** We evaluate our approach on four real-world datasets: *Twitter-LDL* [61] (10,045 Twitter images labeled for eight emotions), *Flickr-LDL* [61] (10,700 Flickr images annotated for eight emotions by 11 annotators), *Emotion6* [43] (1,980 Flickr images labeled for six emotions), and *RAF-LDL* [34] (5,000 multi-label facial expression images).

**Implementation** Following [5, 58, 65], we employ ResNet-50 [16, 44, 55] pre-trained on ImageNet [32, 57] for training the classification model. For training images, we adopt standard flip-and-shift strategy [52, 51, 64] for weak data augmentation, and RandAugment [7, 46] and Cutout [8, 14, 3] for strong data augmentation. We employ AdamW [62, 45] optimizer and one-cycle policy scheduler [15] to train the model with maximal learning rate of 0.0001. For all datasets, the number of epochs is set as 30 and the batch size is set as 32. Furthermore, we perform exponential moving average (EMA) [25, 40] for the model parameter $\theta$ with a decay of 0.98. We adjust the parameter $\lambda$ across a range of values, specifically $\{0.005, 0.01, 0.05, 0.1\}$. We perform all experiments on GeForce RTX 3090 GPUs. The random seed is set to 1 for all experiments. The datasets detail and the other implementation detail can be find in Appendix C.

**Comparing Methods.** To evaluate the effectiveness of our proposed RankMatch method, we benchmark it against four distinct groups of algorithms:

- *Semi-Supervised Multi-Label Learning (SSMLL) Algorithms:* We introduce two advanced algorithms, SSMLL-CAP(CAP) [59] and PCLP [36], developed to address the challenges of semi-supervised multi-label learning by improving the reliability of pseudo-labeling and leveraging label correlations within multi-label datasets.

Table 1: Comparison of testing results on the Emotion6, Flickr, RAF, and Twitter datasets using Canberra, Clark, Intersection, and Cosine metrics. The table reports performance under different labeled data proportions (10%, 20%, and 40%) used for training. The best performance in each metric is highlighted in bold.

| | | Emotion6 | | | Flickr-LDL | | | Twitter-LDL | | | RAF-LDL | | |
|---|---|---|---|---|---|---|---|---|---|---|---|---|---|
| | Method | 10% | 20% | 40% | 10% | 20% | 40% | 10% | 20% | 40% | 10% | 20% | 40% |
| Can.↓ | Rankmatch | **3.3902** | **3.3176** | 3.2504 | **4.4060** | **3.9964** | **3.9013** | **3.7370** | **3.6962** | 3.2913 | **3.0178** | **2.9358** | **2.8341** |
| | SSMLL-CAP | 3.7951 | 3.7613 | 3.7248 | 5.3827 | 5.3235 | 5.2676 | 5.8983 | 5.7659 | 5.6366 | 3.4385 | 3.2808 | 3.1966 |
| | PCLP | 3.7011 | 3.6017 | 3.6030 | 5.2781 | 5.2292 | 5.1966 | 5.4909 | 5.3738 | 5.4133 | 3.3696 | 3.3383 | 3.3310 |
| | Fixmatch-LDL | 3.5080 | 3.5680 | 3.6050 | 5.5570 | 5.5310 | 5.4350 | 6.1750 | 6.0060 | 5.8340 | 3.1220 | 3.0920 | 3.0770 |
| | Mixmatch-LDL | 3.6080 | 3.4860 | 3.4880 | 5.6450 | 5.5026 | 5.5750 | 6.3530 | 6.2489 | 6.2960 | 3.1580 | 3.1111 | 3.0630 |
| | GCT-LDL | 3.5980 | 3.5490 | 3.6410 | 5.5860 | 5.5872 | 5.5260 | 6.3010 | 6.3078 | 6.2380 | 3.1920 | 3.1260 | 3.1470 |
| | SALDL | 3.4836 | 3.3737 | **3.1931** | 5.4612 | 4.7789 | 4.8199 | 5.0380 | 4.0868 | 4.0742 | 3.1947 | 3.1415 | 3.0527 |
| | sLDLF | 4.4164 | 4.3398 | 4.1322 | 6.2280 | 6.1238 | 6.2589 | 5.3084 | 6.0008 | 6.1910 | 4.0586 | 4.1705 | 4.1189 |
| | DF-LDL | 4.2427 | 4.0717 | 3.7221 | 5.5348 | 5.5549 | 5.5207 | 6.4184 | 6.3120 | 6.2588 | 3.3281 | 3.3865 | 3.3582 |
| | LDL-LRR | 4.6528 | 4.0496 | 3.7719 | 5.6325 | 5.4988 | 5.4319 | 6.4215 | 6.3295 | 6.2905 | 3.8677 | 4.0116 | 4.1890 |
| | Adam-LDL-SCL | 4.0815 | 4.1128 | 4.1204 | 6.1634 | 5.9889 | 5.6508 | 6.5220 | 6.4081 | 6.3575 | 3.0891 | 3.0242 | 2.9912 |
| Cla.↓ | Rankmatch | **1.5298** | **1.5050** | **1.4834** | **1.8189** | **1.7051** | **1.6737** | **1.6480** | **1.6190** | **1.5138** | **1.4506** | **1.4190** | **1.3843** |
| | SSMLL-CAP | 1.6705 | 1.6611 | 1.6502 | 2.1222 | 2.0988 | 2.0820 | 2.2590 | 2.2155 | 2.1733 | 1.5918 | 1.5332 | 1.5082 |
| | PCLP | 1.6397 | 1.6059 | 1.6083 | 2.0601 | 2.0478 | 2.0328 | 2.1002 | 2.0623 | 2.0728 | 1.5689 | 1.5636 | 1.5593 |
| | Fixmatch-LDL | 1.5950 | 1.6230 | 1.6390 | 2.2220 | 2.2110 | 2.1910 | 2.3830 | 2.3310 | 2.2820 | 1.5130 | 1.5060 | 1.5050 |
| | Mixmatch-LDL | 1.6240 | 1.5810 | 1.5840 | 2.2330 | 2.1996 | 2.2160 | 2.4280 | 2.4034 | 2.4150 | 1.5150 | 1.5020 | 1.4870 |
| | GCT-LDL | 1.6090 | 1.6050 | 1.6390 | 2.2200 | 2.2238 | 2.2080 | 2.4170 | 2.4216 | 2.4060 | 1.5350 | 1.5170 | 1.5290 |
| | SALDL | 1.6019 | 1.5751 | 1.5100 | 2.1967 | 2.0369 | 2.0446 | 2.1288 | 1.8938 | 1.8964 | 1.5445 | 1.5288 | 1.5035 |
| | sLDLF | 1.8922 | 1.8566 | 1.8049 | 2.3722 | 2.3436 | 2.3761 | 2.1480 | 2.3384 | 2.3746 | 1.9300 | 1.9645 | 1.9750 |
| | DF-LDL | 1.8217 | 1.7746 | 1.6781 | 2.2253 | 2.2072 | 2.1992 | 2.4313 | 2.4108 | 2.4033 | 1.6071 | 1.6229 | 1.6138 |
| | LDL-LRR | 1.9899 | 1.7745 | 1.6953 | 2.2285 | 2.2026 | 2.1919 | 2.4429 | 2.4223 | 2.4121 | 1.7907 | 1.8298 | 1.8919 |
| | Adam-LDL-SCL | 1.7851 | 1.7976 | 1.8014 | 2.3534 | 2.3093 | 2.2312 | 2.4639 | 2.4324 | 2.4160 | 1.5134 | 1.4980 | 1.4905 |
| Int.↑ | Rankmatch | **0.6735** | **0.6832** | **0.6940** | **0.6921** | **0.7073** | **0.7151** | **0.7036** | **0.7190** | **0.7316** | 0.6551 | **0.6813** | **0.7044** |
| | SSMLL-CAP | 0.5479 | 0.5587 | 0.5666 | 0.5815 | 0.6125 | 0.6377 | 0.6034 | 0.6324 | 0.6577 | 0.5264 | 0.5876 | 0.6092 |
| | PCLP | 0.6059 | 0.6370 | 0.6363 | 0.6392 | 0.6469 | 0.6490 | 0.6707 | 0.6784 | 0.6780 | 0.5471 | 0.5588 | 0.5590 |
| | fixmatch-LDL | 0.6638 | 0.6797 | 0.6916 | 0.6857 | 0.7042 | 0.7119 | 0.7009 | 0.7147 | 0.7283 | **0.6570** | 0.6760 | 0.6987 |
| | Mixmatch-LDL | 0.6372 | 0.6418 | 0.6496 | 0.6639 | 0.6686 | 0.6831 | 0.6819 | 0.6806 | 0.6986 | 0.6133 | 0.6381 | 0.6534 |
| | GCT-LDL | 0.6116 | 0.6602 | 0.6770 | 0.6639 | 0.6879 | 0.6863 | 0.6787 | 0.7018 | 0.7102 | 0.6321 | 0.6669 | 0.6910 |
| | SALDL | 0.6457 | 0.6612 | 0.6723 | 0.5559 | 0.5108 | 0.5091 | 0.6632 | 0.5724 | 0.5687 | 0.6298 | 0.6504 | 0.6708 |
| | sLDLF | 0.5935 | 0.5861 | 0.6162 | 0.4813 | 0.4750 | 0.4616 | 0.6487 | 0.5652 | 0.5336 | 0.2433 | 0.2315 | 0.2199 |
| | DF-LDL | 0.5057 | 0.5461 | 0.6353 | 0.4173 | 0.4176 | 0.4169 | 0.3541 | 0.3536 | 0.3505 | 0.7022 | 0.7083 | 0.7085 |
| | LDL-LRR | 0.3721 | 0.6213 | 0.6626 | 0.5322 | 0.5519 | 0.5600 | 0.5746 | 0.5904 | 0.5979 | 0.5649 | 0.5389 | 0.4411 |
| | Adam-LDL-SCL | 0.3409 | 0.5627 | 0.6040 | 0.4724 | 0.3933 | 0.4628 | 0.5488 | 0.5828 | 0.5200 | 0.6177 | 0.5768 | 0.4843 |
| Cos.↑ | Rankmatch | **0.8121** | **0.8257** | **0.8331** | **0.8489** | **0.8614** | **0.8679** | **0.8544** | **0.8698** | **0.8790** | **0.7901** | **0.8140** | **0.8375** |
| | SSMLL-CAP | 0.6850 | 0.6994 | 0.7185 | 0.7634 | 0.7885 | 0.8144 | 0.8109 | 0.8270 | 0.8442 | 0.6456 | 0.7119 | 0.7329 |
| | PCLP | 0.7421 | 0.7737 | 0.7778 | 0.8057 | 0.8146 | 0.8151 | 0.8391 | 0.8436 | 0.8448 | 0.6815 | 0.6962 | 0.6969 |
| | Fixmatch-LDL | 0.8079 | 0.8200 | 0.8312 | 0.8487 | 0.8573 | 0.8673 | 0.8517 | 0.8647 | 0.8758 | 0.7881 | 0.8123 | 0.8311 |
| | Mixmatch-LDL | 0.7585 | 0.7863 | 0.7901 | 0.7888 | 0.8381 | 0.8468 | 0.8463 | 0.8552 | 0.8602 | 0.7536 | 0.7680 | 0.7820 |
| | GCT-LDL | 0.7530 | 0.8017 | 0.8134 | 0.8313 | 0.8508 | 0.8531 | 0.8499 | 0.8587 | 0.8716 | 0.7660 | 0.7977 | 0.8181 |
| | SALDL | 0.7784 | 0.7874 | 0.7981 | 0.7361 | 0.6643 | 0.6624 | 0.8479 | 0.7612 | 0.7615 | 0.7711 | 0.7938 | 0.8135 |
| | sLDLF | 0.7037 | 0.6980 | 0.7350 | 0.6276 | 0.6066 | 0.5897 | 0.8002 | 0.7454 | 0.6988 | 0.3262 | 0.3506 | 0.3459 |
| | DF-LDL | 0.6035 | 0.6470 | 0.7689 | 0.5436 | 0.5539 | 0.5569 | 0.5069 | 0.5233 | 0.5209 | 0.8427 | 0.8492 | 0.8470 |
| | LDL-LRR | 0.4604 | 0.7362 | 0.7905 | 0.7020 | 0.7316 | 0.7399 | 0.7767 | 0.8027 | 0.8125 | 0.7253 | 0.6938 | 0.5757 |
| | Adam-LDL-SCL | 0.4311 | 0.6670 | 0.7144 | 0.6104 | 0.4888 | 0.6166 | 0.7163 | 0.7661 | 0.7403 | 0.7717 | 0.7337 | 0.6191 |

Table 2: Evaluation of Label Distribution Ranking Relationships for Test Samples. The table compares Kendall tau ($\tau_K$) and Spearmans rank ($\rho_S$) correlation coefficients across datasets and different methods. Bold values indicate the best-performing method for each dataset and metric.

| Metric | Dataset | 10% Labeled Data | | | | | | 20% Labeled Data | | | | | |
|---|---|---|---|---|---|---|---|---|---|---|---|---|---|
| | | RankMatch | GCT | CAP | PCLP | FixMatch | MixMatch | RankMatch | GCT | CAP | PCLP | FixMatch | MixMatch |
| $\tau_K$ | RAF | **0.5696** | 0.4258 | 0.2079 | 0.2750 | 0.4643 | 0.4066 | **0.5463** | 0.4811 | 0.3598 | 0.2949 | 0.4946 | 0.4524 |
| | Emotion6 | **0.5535** | 0.3718 | 0.1237 | 0.3325 | 0.4899 | 0.4562 | **0.5620** | 0.4594 | 0.1536 | 0.4394 | 0.4985 | 0.4528 |
| | Flickr | **0.5618** | 0.5005 | 0.4030 | 0.4904 | 0.5215 | 0.5119 | **0.5627** | 0.5215 | 0.4475 | 0.5005 | 0.5416 | 0.5265 |
| | Twitter | 0.4927 | 0.4806 | 0.4407 | 0.4887 | 0.4744 | **0.5016** | **0.5452** | 0.5121 | 0.4828 | 0.5037 | 0.5039 | 0.5177 |
| $\rho_S$ | RAF | **0.6726** | 0.5122 | 0.2474 | 0.3365 | 0.5564 | 0.4900 | **0.6649** | 0.5754 | 0.4297 | 0.3602 | 0.5917 | 0.5497 |
| | Emotion6 | **0.6545** | 0.4495 | 0.1529 | 0.4037 | 0.5933 | 0.5528 | **0.6512** | 0.5624 | 0.1923 | 0.5332 | 0.5989 | 0.5559 |
| | Flickr | **0.6537** | 0.5904 | 0.4831 | 0.5793 | 0.6097 | 0.6019 | **0.6551** | 0.6112 | 0.5335 | 0.5903 | 0.6304 | 0.6171 |
| | Twitter | 0.5740 | 0.5637 | 0.5193 | 0.5735 | 0.5517 | **0.5864** | **0.6315** | 0.5966 | 0.5658 | 0.5898 | 0.5853 | 0.6024 |

- *Dual-Network SSLDL Algorithm:* We present and evaluate our own GCT-LDL(GCT), a dual-network [4] SSLDL approach that we developed, which leverages mutual supervision of unlabeled data between two independent networks for enhanced learning.

- *Deep Learning SSLDL Algorithms:* We introduce two novel algorithms, FixMatch-LDL [52] and MixMatch-LDL [2], designed to bridge the gap in open-source semi-supervised LDL (SSLDL) approaches within deep learning frameworks.

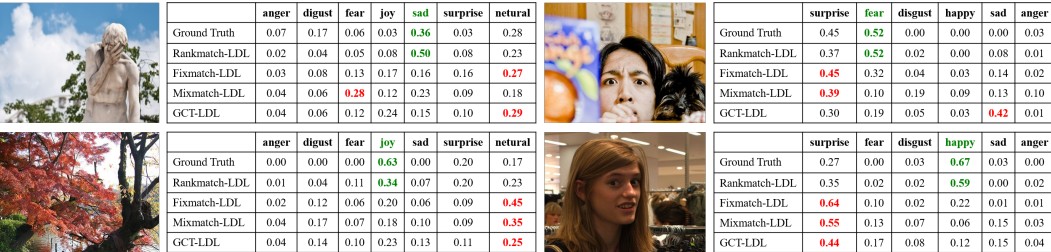

Figure 3: Examples illustrating *RankMatch-LDL*'s ability to generate superior pseudo-label distributions compared to existing semi-supervised methods. The first two images are from the Emotion6 [43] dataset, and the last two are from the RAF dataset [34]. *RankMatch-LDL* more accurately aligns with the ground truth label ranking.

- *Traditional SSLDL Algorithm:* The traditional SA-LDL [18] algorithm, originally for tabular data, is adapted for image datasets through necessary feature engineering, detailed in Appendix D.
- *SOTA LDL Algorithms:* Comparisons are also made with state-of-the-art LDL algorithms including Adam-LDL-SCL [20], sLDLF [50], DF-LDL [13], and LDL-LRR [21], highlighting their potential limitations in SSLDL contexts.

**Evaluation Metrics.** We evaluate LDL methods using eight metrics [10]: Chebyshev, Clark, Canberra distances, and Kullback-Leibler divergence (lower is better), along with Intersection and Cosine similarities, Spearmans rank correlation ($\rho_S$), and Kendall tau correlation ($\tau_K$)[21] (higher is better).

## 6.1 Comparative Experiment Analysis

We employed a range of labeled data proportions (10%, 20%, and 40%) to simulate varying levels of label availability, a critical factor in semi-supervised learning scenarios. The experiments are presented in Table 1 and Table 2. from that we can draw the following conclusions

- RankMatch consistently achieves top performance across all datasets and metrics. Compared to SSMLL-CAP and PCLP, RankMatch shows stronger label relationship modeling, leveraging PRR losses to refine pseudo-label rankings and outperforming traditional SSLDL methods like FixMatch-LDL and MixMatch-LDL.
- The results in Table 2 confirm the consistent superiority of *RankMatch* in capturing label ranking relationships, achieving the highest Kendall tau ($\tau_K$) and Spearmans rank ($\rho_S$) correlation coefficients across almost all datasets and metrics. Notably, *RankMatch* maintains robust performance as the proportion of labeled data increases, demonstrating strong generalization and adaptability under varying supervision levels.
- With increasing labeled data, RankMatch scales effectively, achieving significant performance improvements. On Twitter, the Canberra distance improves from 3.7370 (10% labeled data) to 3.2913 (40% labeled data).

## 6.2 Analysis of Pseudo-Label Performance

To evaluate the performance of pseudo-labeling, we assess the pseudo-labeling quality of different algorithms and visualize two sample images from the Emotion6 and RAF datasets as examples. The experimental results are presented in Table 3 and Fig. 3. Based on these results, we derive the following conclusions:

- Our method effectively captures the true label distribution, achieving the best performance in overall distance metrics, as evidenced by the lowest KL divergence across all datasets.
- By incorporating the PRR loss, our approach generates pseudo-labels with ranking structures that more closely align with the ground truth, leading to higher-quality pseudo-labels and improved model performance.

Table 3: Comparison of pseudo-labeling performance with different methods trained on 10% and 20% labeled data. Kendalls Tau ($\tau_K$) and Spearmans rank correlation ($\rho_S$) measure ranking quality (higher is better), while KL divergence quantifies distribution alignment (lower is better).

| Dataset | 10% Labeled Data | | | | | | 20% Labeled Data | | | | | |
|---|---|---|---|---|---|---|---|---|---|---|---|---|
| | RankMatch | FixMatch | MixMatch | GCT | CAP | PCLP | RankMatch | FixMatch | MixMatch | GCT | CAP | PCLP |
| | | | | | | $\rho_S$ | | | | | | |
| Emotion6 | **0.6266** | 0.5372 | 0.4763 | 0.3470 | 0.1838 | 0.2921 | **0.6287** | 0.5685 | 0.4760 | 0.4929 | 0.4449 | 0.3325 |
| RAF | **0.6617** | 0.5348 | 0.5024 | 0.4774 | 0.2593 | 0.3471 | **0.6730** | 0.5772 | 0.5480 | 0.5730 | 0.2354 | 0.4313 |
| Flickr | **0.6646** | 0.6262 | 0.6153 | 0.6024 | 0.4902 | 0.5933 | **0.6618** | 0.6490 | 0.6377 | 0.6328 | 0.5528 | 0.5988 |
| Twitter | **0.6446** | 0.5551 | 0.6006 | 0.5729 | 0.5197 | 0.5681 | **0.6440** | 0.5990 | 0.6147 | 0.6108 | 0.5602 | 0.5830 |
| | | | | | | KL | | | | | | |
| Emotion6 | **2.5458** | 3.4885 | 3.9461 | 5.1156 | 5.6353 | 5.0965 | **2.4344** | 3.0090 | 4.1522 | 3.7812 | 3.5549 | 4.8452 |
| RAF | **2.0564** | 2.9951 | 3.9584 | 3.5445 | 4.9070 | 4.5420 | **1.9380** | 2.5717 | 3.1612 | 2.7052 | 5.4632 | 3.9629 |
| Flickr | **2.6495** | 3.7454 | 3.6983 | 4.2128 | 6.0978 | 3.7667 | **2.6802** | 2.9860 | 4.0924 | 3.7720 | 5.2461 | 3.7492 |
| Twitter | **2.5243** | 3.1827 | 3.4360 | 4.1143 | 6.5147 | 3.2599 | **2.5900** | 3.1581 | 3.9771 | 3.2475 | 5.7258 | 3.0865 |

Table 4: Ablation Results on Flickr and RAF Datasets.

| | | Che.↓ | Cla.↓ | Can.↓ | KL↓ | Cos.↑ | Int.↑ |
|---|---|---|---|---|---|---|---|
| | pretrain | 0.2411 | 2.2594 | 5.6885 | 0.5371 | 0.8427 | 0.6873 |
| Flickr | pretrain + consistency | 0.2262(6.2%↑) | 2.1131(6.5%↑) | 5.1536(9.4%↑) | 0.5293(1.5%↑) | 0.8633(2.4%↑) | 0.7188(4.6%↑) |
| | pretrain + consistency+PRR loss | **0.2184(3.4%↑)** | **2.0158(4.6%↑)** | **4.9008(4.9%↑)** | **0.5227(1.2%↑)** | **0.8714(0.9%↑)** | **0.7208(0.3%↑)** |
| | | Che.↓ | Cla.↓ | Can.↓ | KL↓ | Cos.↑ | Int.↑ |
| | pretrain | 0.2938 | 1.5412 | 3.206 | 0.5146 | 0.7687 | 0.6411 |
| RAF | pretrain + consistency | 0.255(13.2%↑) | 1.5021(2.5%↑) | 3.1345(2.2%↑) | 0.3699(28.1%↑) | 0.8189(28.1%↑) | 0.7073(10.3%↑) |
| | pretrain + consistency+PRR loss | **0.2341(8.2%↑)** | **1.4914(0.7%↑)** | **3.0459(2.8%↑)** | **0.3464(6.4%↑)** | **0.8476(3.5%↑)** | **0.7194(1.7%↑)** |

## 6.3 Further Analysis

**Ablation Study** Our ablation study analyzed the impact of PRR loss and unsupervised consistency loss on the performance of RankMatch. Initially, the model was pre-trained with only 10% of labeled data to establish a baseline. This phase highlighted the model's ability to utilize minimal data effectively.

Next, unsupervised consistency loss was applied to enhance learning from unlabeled data. In the final phase, PRR loss was introduced, leveraging the same 10% labeled data to refine the model further with supervised ranking loss. Ablation experiment results are shown in Table 4. From this, we can draw the following conclusions

- The integration of unsupervised consistency loss markedly improves RankMatch's performance across datasets, as observed in the ablation results. This confirms the effectiveness of using unsupervised data to enhance model accuracy.

- The incorporation of pairwise relevance ranking (PRR) loss significantly boosts performance, particularly in scenarios where it surpasses the baseline. This improvement demonstrates the PRR loss's critical role in refining label discrimination within the semi-supervised learning framework.

**The Impact of $\mathcal{L}_{PRR}$ Loss on Pseudo-Labeling**: We evaluated the effect of $\mathcal{L}_{PRR}$ loss on pseudo-labeling by comparing models trained with and without it on the RAF and Emotion6 datasets. Pseudo-labels were generated for part of the validation set during training, and their ranking performance was analyzed. As shown in Fig. 4, incorporating PRR loss consistently improved the ranking quality of pseudo-labels, aligning them more closely with the ground truth. This demonstrates the ability of $\mathcal{L}_{PRR}$ loss to effectively capture inter-label ranking relationships, thereby enhancing both the training process and the overall model performance.

**Parameter Sensitivity Analysis** Fig. 5 illustrates the impact of the parameter $\lambda$ on RankMatch's performance across the Emotion6, Flickr-LDL, RAF-LDL, and Twitter-LDL datasets, focusing on KL divergence and Cosine similarity metrics. Fig. 5 shows that RankMatch performs consistently well when $\lambda$ ranges from 0.01 to 0.05, with minimal variations in performance metrics such as KL

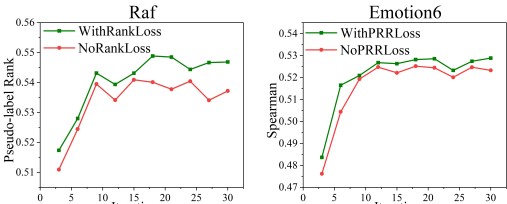

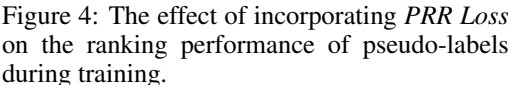

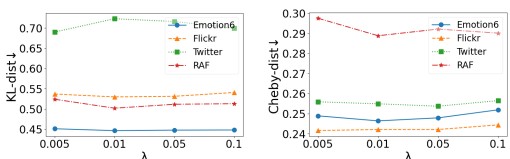

Figure 4: The effect of incorporating *PRR Loss* on the ranking performance of pseudo-labels during training.

Figure 5: Parameter sensitivity analysis on Emotion6, Flickr-LDL, RAF-LDL, and Twitter-LDL datasets.

divergence, Intersection, and Cosine similarity across all datasets. However, at $\lambda = 0.005$, a noticeable performance drop, particularly in the Emotion6 dataset, highlights the reduced effectiveness of regularization. Conversely, higher values, such as $\lambda = 0.1$, slightly hinder performance, suggesting over-regularization. This analysis indicates that $\lambda$ values between 0.01 and 0.05 strike the best balance for effective learning. Detailed results for additional metrics can be found in Appendix D.

## 7    Conclusion and Limitations

**Conclusion.** In this paper, we introduce RankMatch, the first deep-learningbased semi-supervised label distribution learning (SSLDL) method that explicitly models inter-label ranking relationships via pseudo-labels. By combining the standard KL-divergence loss with our novel Pairwise Ranking Relationship (PRR) loss within a single training framework, RankMatch produces higher-quality pseudo-label distributions and flexibly leverages unlabeled data without overfitting to noisy absolute values. In our theoretical analysis, we prove that adding the PRR loss tightens the models generalization bound and provide an explicit generalization error bound for RankMatch. Finally, we empirically validate on four real-world LDL benchmarks (Twitter-LDL, Flickr-LDL, Emotion6, and RAF-LDL) that RankMatch consistently outperforms all baselines.

**Limitations.** While SSLDL significantly lowers the demand for fully annotated data, it still depends on a nontrivial amount of manual labeling. Going forward, we plan to investigate how to harness large language models to generate cost-effective, high-quality pseudo-labels and to integrate LLMs directly into the SSLDL training pipeline.

## Acknowledgments

This work was supported in part by the Jiangsu Science Foundation (BG2024036, BK20243012), the National Natural Science Foundation of China (62125602, U24A20324, 92464301, 62306073), the China Postdoctoral Science Foundation (2022M720028), the Xplorer Prize, and the National Natural Science Foundation of China under Grant U24A20322. Additional support was provided by the Big Data Computing Center of Southeast University.

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

## A  The Proof of Theorem 3.1

We now show that adding the pairwise relevance ranking (PRR) loss to the usual KLdivergence objective strictly tightens the generalization bound in semisupervised label distribution learning.

**Theorem A.1.** *Let $\mathcal{F}$ be a hypothesis class of scoring functions $f\colon \mathcal{X} \to \mathbb{R}^C$. Define the empirical risks on the labeled set $\{(x_i, d_i)\}_{i=1}^n$ and unlabeled set $\{x_j\}_{j=1}^m$ by*

$$\widehat{R}_{\mathrm{KL}}(f) = \frac{1}{n}\sum_{i=1}^n D_{\mathrm{KL}}\big(d_i \,\|\, f(x_i)\big),\ \widehat{R}_{\mathrm{PRR}}(f) = \frac{1}{m}\sum_{j=1}^m L_{\mathrm{PRR}}\big(f(x_j), \hat{d}_j\big),$$

*where $D_{\mathrm{KL}}(\cdot\|\cdot)$ is the KullbackLeibler divergence and $L_{\mathrm{PRR}}$ is the pairwise ranking loss computed against pseudo-labels $\hat{d}_j$. Let the combined empirical risk be*

$$\widehat{R}_{\mathrm{tot}}(f) = \widehat{R}_{\mathrm{KL}}(f) \ + \ \lambda\, \widehat{R}_{\mathrm{PRR}}(f).$$

*Denote*

$$f_{\mathrm{KL}} = \arg\min_{f\in\mathcal{F}} \widehat{R}_{\mathrm{KL}}(f), \qquad f_{\mathrm{KL+PRR}} = \arg\min_{f\in\mathcal{F}} \widehat{R}_{\mathrm{tot}}(f).$$

*Assume all losses are bounded by $B$ and Lipschitz continuous. Define the population risks*

$$R_{\mathrm{KL}}(f) = \mathbb{E}\big[D_{\mathrm{KL}}(d\|f(x))\big], \quad R_{\mathrm{PRR}}(f) = \mathbb{E}\big[L_{\mathrm{PRR}}(f(x), d)\big], \quad R_{\mathrm{tot}}(f) = R_{\mathrm{KL}}(f) + \lambda\, R_{\mathrm{PRR}}(f).$$

*Then for any $\delta \in (0, 1)$, with probability at least $1 - \delta$ over the training draw,*

$$R_{\mathrm{tot}}\big(f_{\mathrm{KL+PRR}}\big) \ < \ R_{\mathrm{tot}}\big(f_{\mathrm{KL}}\big) \ + \ 2\,\mathfrak{R}_{n+m}\big(\ell \circ \mathcal{F}\big) \ + \ B\sqrt{\frac{\ln(1/\delta)}{2(n+m)}},$$

*where $\mathfrak{R}_{n+m}(\ell\circ\mathcal{F})$ is the Rademacher complexity of the combined loss class $\ell(x, y) = D_{\mathrm{KL}}(y\|x) + \lambda L_{\mathrm{PRR}}(x, y)$. In particular, if adding $L_{\mathrm{PRR}}$ strictly lowers the empirical total risk, then the resulting population total risk is strictly smaller than that of the KL-only solution.*

*Proof.* By standard Rademachercomplexity bounds (see [41]), for every $f \in \mathcal{F}$, with probability at least $1 - \delta$,

$$R_{\mathrm{tot}}(f) \ \leq \ \widehat{R}_{\mathrm{tot}}(f) \ + \ 2\,\mathfrak{R}_{n+m}(\ell \circ \mathcal{F}) \ + \ B\sqrt{\tfrac{\ln(1/\delta)}{2(n+m)}}.$$

By definition of the minimizers,

$$\widehat{R}_{\mathrm{KL}}\big(f_{\mathrm{KL}}\big) \ \leq \ \widehat{R}_{\mathrm{KL}}\big(f_{\mathrm{KL+PRR}}\big), \quad \widehat{R}_{\mathrm{tot}}\big(f_{\mathrm{KL+PRR}}\big) \ \leq \ \widehat{R}_{\mathrm{tot}}\big(f_{\mathrm{KL}}\big).$$

Subtracting these two inequalities shows that

$$\widehat{R}_{\mathrm{tot}}\big(f_{\mathrm{KL+PRR}}\big) - \widehat{R}_{\mathrm{tot}}\big(f_{\mathrm{KL}}\big) \ < \ 0 \quad \Longrightarrow \quad \widehat{R}_{\mathrm{tot}}\big(f_{\mathrm{KL+PRR}}\big) < \widehat{R}_{\mathrm{tot}}\big(f_{\mathrm{KL}}\big).$$

Apply the uniformconvergence bound from Step 1 to both $f_{\mathrm{KL+PRR}}$ and $f_{\mathrm{KL}}$. The difference in their population risks is upperbounded by the difference in empirical risks (which is negative by Step 2) plus the same complexity term. Hence

$$R_{\mathrm{tot}}\big(f_{\mathrm{KL+PRR}}\big) < R_{\mathrm{tot}}\big(f_{\mathrm{KL}}\big) + 2\,\mathfrak{R}_{n+m}(\ell \circ \mathcal{F}) + B\sqrt{\tfrac{\ln(1/\delta)}{2(n+m)}},$$

as claimed. $\qquad\qquad\qquad\qquad\qquad\qquad\qquad\qquad\qquad\qquad\qquad\qquad\qquad\qquad\square$

## B  The Proof of the Theorem 3.2

We study the generalization performance of Rankmatch. Before providing the main results, we first define the true risk with respect to the classification model $f(x; \theta)$:

$$R(f) = \mathbb{E}_{(x,y)}[L(f(\mathbf{x}), \mathbf{d})].$$

Our goal is to learn a good classification model by minimizing the empirical risk $\hat{R}(f) = \hat{R}_L(f) + \hat{R}_U(f)$, where $\hat{R}_L(f)$ and $\hat{R}_U(f)$ are respectively the empirical risk of the labeled loss $L_L(f(\mathbf{x}), \mathbf{d})$ and unlabeled loss $L_U(f(\mathbf{x}), \mathbf{d})$:

$$\hat{R}_L(f) = \frac{1}{n} \sum_{i=1}^{n} L(f(\mathbf{x}_i), \mathbf{d}_i), \quad \hat{R}_U(f) = \frac{1}{m} \sum_{j=1}^{m} L_U(f(\mathbf{x}_j), \mathbf{d}_j).$$

Note that during the training, we cannot train a model directly by optimizing $\hat{R}_U(f)$, since the labels of unlabeled data are inaccessible. Instead, we train the model with $\hat{R}'_U(f) = \frac{1}{m} \sum_{j=1}^{m} L_U(f(\mathbf{x}_j), \hat{\mathbf{d}}_j)$, where $\hat{\mathbf{d}}_j$ represents the pseudo-label vector of the instance $\mathbf{x}_j$.

Let $L_k(f(\mathbf{x})) = d_{\mathbf{x}}^{y_k} \ln \left( \frac{d_{\mathbf{x}}^{y_k}}{h(y_k | \text{Aug}_w(\mathbf{x}))} \right)$ be the loss for the label k, and $L_E$ be any (not necessarily the best) Lipschitz constant of $L$. Let $R_N(\mathcal{F})$ be the expected Rademacher complexity of $\mathcal{F}$ with $N = m + n$ training points. Let $\hat{f}$ be the empirical risk minimizer, where $\mathcal{F}$ is a function class, and $f^*$ be the true minimizer. We derive the following theorem, which provides a generalization error bound for the proposed method.

**Theorem B.1.** *Suppose that $\ell(\cdot)$ is bounded by B. For some $\epsilon > 0$, if $\sum_{j=1}^{m} | \mathbb{I}(f_k(\mathbf{x}_j)) - \mathbb{I}\left(d_{\mathbf{x}_j}^{y_k}\right) | /m \leq \epsilon$ for any $k \in [q]$, for any $\delta > 0$, with probability at least $1 - \delta$, we have*

$$R(\hat{f}) - R(f^*) \leq 2qB\epsilon + 4qL_E R_N(\mathcal{F}) + 2qB\sqrt{\frac{\log \frac{2}{\delta}}{2N}}.$$

From Theorem 2, it can be observed that the generalization performance of $\hat{f}$ mainly depends on two factors, i.e., the pseudo-labeling error $\epsilon$ and the number of training examples $N$. Apparently, a smaller pseudo-labeling error $\epsilon$ often leads to better generalization performance. Thanks to its robustness and the empirical evidence supporting the model, we anticipate strong performance in practical applications.

**Theorem B.2.** *Suppose that $\ell(\cdot)$ is bounded by B. For some $\epsilon > 0$, if $\sum_{j=1}^{m} | \mathbb{I}(f_k(\mathbf{x}_j)) - \mathbb{I}\left(d_{\mathbf{x}_j}^{y_k}\right) | /m \leq \epsilon$ for any $k \in [q]$ for any $\delta > 0$, with probability at least $1 - \delta$, we have*

$$R(\hat{f}) - R(f^*) \leq 2qB\epsilon + 4qL_E R_N(\mathcal{F}) + 2qB\sqrt{\frac{\log \frac{2}{\delta}}{2N}}.$$

**Proof.** Before proving the theorem, we first provide two useful lemmas as follows. We primarily derive the uniform deviation bound between $R(\hat{f})$ and $R(f)$.

**Lemma B.3.** *Suppose that the loss function $\ell$ is $L_E$-Lipschitz continuous with respect to $\theta$. For any $\delta > 0$, with probability at least $1 - \delta$, we have*

$$|R(\hat{f}) - \hat{R}(f)| \leq 2qL_E R_{n+m}(\mathcal{F}) + qB\sqrt{\frac{\log \frac{2}{\delta}}{2(n+m)}} \tag{12}$$

*Proof.* In order to prove this lemma, we define the Rademacher complexity of $L$ and $\mathcal{F}$ with $m + n$ training examples as follows:

$$R_{n+m}(L \circ \mathcal{F}) = \mathbb{E}_{\mathbf{x}, \mathbf{d}, \sigma} \left[ \sup_{f \in \mathcal{F}} \sum_{i=1}^{n} \sigma_i \ell\left(f\left(\mathbf{x}_i\right), \mathbf{d}_i\right) + \sum_{j=1}^{m} \sigma_j \ell\left(f\left(\mathbf{x}_j\right), \mathbf{d}_j\right) \right]$$

where $\sigma_i$ and $\sigma_j$ are Rademacher variables.

Considering that $C(f(\mathbf{x}), \mathbf{d}) = \sum_{i=1}^{m} \ell(f_k, \mathbf{d}_k)$, we have

$$R_{n+m}(L \circ \mathcal{F}) \leq qR_{n+m}(\ell \circ \mathcal{F}) \leq qL_E R_{n+m}(\mathcal{F})$$

where the second line is due to the Lipschitz continuity of the loss function $\ell$.

Then, we proceed the proof by showing that one direction $\sup_{f \in \mathcal{F}} R(f) - R(\hat{f})$ is bounded with probability at least $1 - \delta/2$, and the other direction can be proved similarly. According to *McDiarmid's inequality* [6], for any $\delta > 0$, with probability at least $1 - \delta/2$, we have

$$\sup_{f \in \mathcal{F}} R(\hat{f}) - R(f) \leq \sup_{f \in \mathcal{F}} R(\hat{f}) - R(f) + qB\sqrt{\frac{\log \frac{2}{\delta}}{2(n+m)}}$$

According to the result in [41] (Theorem 3.3) that shows $\mathbb{E} \sup_{f \in \mathcal{F}} R(\hat{f}) - R(f) \leq 2R_m(\mathcal{F})$, by further considering the other direction $\sup_{f \in \mathcal{F}} R(f) - R(\hat{f})$, with probability at least $1 - \delta$, we have

$$\sup_{f \in \mathcal{F}} \mid R(\hat{f}) - R(f) \mid \leq 2qL_E R_m(\mathcal{F}) + qB\sqrt{\frac{\log \frac{2}{\delta}}{2n+m}}$$

which completes the proof. $\qquad\square$

Then, we can bound the difference between $R(\hat{f})$ and $R(f)$ as follows:

**Lemma B.4.** *Suppose that $\ell(\cdot)$ is bounded by B. For some $\epsilon > 0$, if $\sum_{j=1}^{m} \mid \mathbb{I}(f_k(\mathbf{x}_j)) - \mathbb{I}\left(d_{\mathbf{x}_j}^{y_k}\right) \mid /m \leq \epsilon$ for any $k \in [q]$ for any $\delta > 0$, we have:*

$$\mid \hat{R}_U(f) - R_U(f) \mid \leq qB\epsilon$$

*Proof.* Without loss of generality, assume that $\epsilon$ is the largest pseudo-labeling error among $q$ classes, i.e., $\epsilon = \max_{k=1}^{q} \sum_{j=1}^{m} \mid \mathbb{I}(f_k(\mathbf{x}_j)) - \mathbb{I}\left(d_{\mathbf{x}_j}^{y_k}\right) \mid /m \leq \epsilon$ for any $k \in [q]$. Obviously, $\epsilon$ consists below pseudo-labeling error:

$$\epsilon = \frac{\sum_{j=1}^{m} \mathbb{I}\left(f_k(\mathbf{x}_j), d_{\mathbf{x}_j}^{y_k}\right)}{m} \tag{13}$$

Then, we prove the following side, which provide the bounds for $R_U(f)$. Firstly, we prove its upper bound:

$$\begin{aligned}
\widehat{R}'_u(f) &= \frac{1}{m} \sum_{j=1}^{m} \sum_{k=1}^{q} \mathbb{I}(f_k(\mathbf{x}_j)) \ell(f_k(\mathbf{x}_j)) \\
&\leq \frac{1}{m} \sum_{j=1}^{m} \sum_{k=1}^{q} \mathbb{I}\left(d_{\mathbf{x}_j}^{y_k}\right) \ell(f_k(\mathbf{x}_j)) + \mathbb{I}(d_{\mathbf{x}_j}^{y_k}, f_k(\mathbf{x}_j) \; \ell(f_k(\mathbf{x}_j)) \\
&\leq \frac{1}{m} \sum_{j=1}^{m} \mathcal{L}\left(f(\mathbf{x}_j), d_{\mathbf{x}_j}^{y_k}\right) + \epsilon \sum_{k=1}^{q} \ell(f_k(\mathbf{x}_j)) \\
&\leq \widehat{R}_u(f) + qB\epsilon
\end{aligned} \tag{14}$$

where the second line holds based on Eq.(13). Then, we prove its low bound:

$$\begin{aligned}
\widehat{R}'_u(f) &= \frac{1}{m} \sum_{j=1}^{m} \sum_{k=1}^{q} \mathbb{I}(f_k(\mathbf{x}_j)) \ell(f_k(\mathbf{x}_j)) \\
&\geq \frac{1}{m} \sum_{j=1}^{m} \sum_{k=1}^{q} \mathbb{I}\left(d_{\mathbf{x}_j}^{y_k}\right) \ell(f_k(\mathbf{x}_j)) - \mathbb{I}(d_{\mathbf{x}_j}^{y_k}, f_k(\mathbf{x}_j) \; \ell(f_k(\mathbf{x}_j)) \\
&\geq \frac{1}{m} \sum_{j=1}^{m} \mathcal{L}\left(f(\mathbf{x}_j), d_{\mathbf{x}_j}^{y_k}\right) + \epsilon \sum_{k=1}^{q} \ell(f_k(\mathbf{x}_j)) \\
&\geq \widehat{R}_u(f) + qB\epsilon
\end{aligned} \tag{15}$$

By combining these two sides, we can obtain the following result:

$$|\hat{R}_U(f) - R_U(f)| \leq qB\epsilon$$

which concludes the proof.

For any $\delta > 0$, with probability at least $1 - \delta$, we have:

$$R(f) \leq \hat{R}(f) + R_U(f) + 2qL_E R_{n+m}(\mathcal{F}) + qB\sqrt{\frac{\log \frac{2}{\delta}}{2N}}$$

$$\leq \hat{R}(f) + R_U(f) + qB\epsilon + 2qL_E R_{n+m}(\mathcal{F}) + qB\sqrt{\frac{\log \frac{2}{\delta}}{2N}}$$

$$\leq \hat{R}(f) + R_U(f) + 2qB\epsilon + 2qL_E R_{n+m}(\mathcal{F}) + qB\sqrt{\frac{\log \frac{2}{\delta}}{2N}}$$

$$\leq \hat{R}(f) + R_U(f) + 2qB\epsilon + 4qL_E R_{n+m}(\mathcal{F}) + 2qB\sqrt{\frac{\log \frac{2}{\delta}}{2N}}$$

$$\leq R(f) + 2qB\epsilon + 4qL_E R_{n+m}(\mathcal{F}) + 2qB\sqrt{\frac{\log \frac{2}{\delta}}{2N}}$$

where the first and fifth lines are based on Eq. 6, and second and fourth lines are due to Lemma B.3. The third line is by the definition of $f$. Putting all these together, the proof is then finished. □

## C  Others

**Experimental Datasets** :In this paper, we validate our approach using four distinct real-world datasets [3]. The details of these datasets are as follows:

Twitter-LDL : A large-scale Visual Sentiment Distribution dataset was constructed from Twitter, encompassing eight distinct emotions Amusement, Anger, Awe, Contentment, Disgust, Excitement, Fear, Sadness. Approximately 30,000 images were collected by searching various emotional keywords, such as "sadness," "heartbreak," and "grief." Subsequently, eight annotators were hired to label this dataset. The resulting Twitter LDL dataset comprises 10,045 images.

Flickr-LDL : A subset of the Flickr dataset , unlike other datasets that searched for images using emotional terms, the Flickr dataset collected 1,200 pairs of adjective-noun pairs, resulting in 500,000 images. We employed 11 annotators to label this subset with tags for eight common emotions. In the end, the Flickr LDL was created, containing 10,700 images, with roughly equal quantities for each class.

Emotion6 : Emotion6: We collected 1,980 images from Flickr using six category keywords and synonyms as search terms for Emotion6. A total of 330 images were collected for each category, and each image was assigned to only one category (dominant emotion). Emotion6 represents the emotions related to each image in the form of a probability distribution, consisting of 7 bins, including Ekman's 6 basic emotions and neutral.

RAF-LDL : RAF-LDL is a multi-label distribution facial expression dataset, comprising approximately 5,000 diverse facial images downloaded from the internet. These images exhibit variations in emotion, subject identity, head pose, lighting conditions, and occlusions. During annotation, 315 well-trained annotators are employed to ensure each image can be annotated enough independent times. And images with multi-peak label distribution are selected out to constitute the RAF-LDL.

**Comparing methods** In order to assess the effectiveness of the proposed approach, we benchmark it against four sets of methods:

1) For the semi-supervised multi-label learning (SSMLL) algorithms, we follow the hyperparameter configurations provided in their original papers. Specifically, for SSMLL-CAP (CAP) [59] and

---

[3]The dataset's author has made the dataset publicly available at the following link: http://cv.nankai.edu.cn/projects/SentiLDL.

PCLP [36], we adopt the same backbone networks, learning rates, batch sizes, and training schedules as specified in their respective works. Additionally, to adapt these methods to our semi-supervised label distribution learning (SSLDL) setting, we modify their final activation layer: the original sigmoid function is replaced with a softmax activation. This transformation enables the models to generate pseudo-label distributions rather than independent multi-label probabilities, ensuring better

2) The second group consists of two deep learning SSLDL algorithms that we introduced, named FixMatch-LDL and MixMatch-LDL. Since there are currently no open-source semi-supervised LDL works in deep learning, these two algorithms were developed by us, based on the current most effective two deep learning SSL algorithms.

(i) FixMatch-LDL :Fixmatch-LDL is an adaptation we made based on the classic semi-supervised algorithm fixmatch [52]. Specifically, we pre-trained on images using ResNet50, then trained the model with labeled data. Subsequently, we assigned pseudo-label distributions to the unlabeled data, and finally, we aligned the model's strongly augmented output with the pseudo-label distribution. For all datasets, the number of epochs is set as 30 and the batch size is set as 32. We perform all experiments on GeForce RTX 3090 GPUs. The random seed is set to 1 for all experiments.

(ii) MixMatch-LDL: Mixmatch is a semi-supervised LDL algorithm designed by us. Specifically, we first use linear interpolation to blend images, creating new samples. Similarly, we generate the label distributions for these new samples. Following this, we train the data using the same training strategy as Mixmatch. It's worth mentioning that producing new samples enhances the model's ability to prevent overfitting. For all datasets, the number of epochs is set as 30 and the batch size is set as 32. We perform all experiments on GeForce RTX 3090 GPUs. The random seed is set to 1 for all experiments.

3) The thrid group of algorithms is a deep learning SSLDL algorithm based on the dual-network concept, which we named GCT-LDL. The core idea involves mutual supervision of the outputs from two independent networks using unlabeled data. GCT-LDL : Two models utilized two different pretrained initializations of ResNet50 provided by PyTorch (ResNet50-Weights.IMAGENET1K-V1 and ResNet50-Weights.IMAGENET1K-V2). During training, labeled and unlabeled data were mixed. The loss used is the cross-entropy loss, divided into two parts: for labeled data, the loss is calculated directly between the prediction results and the ground truth. For unlabeled data, the loss is calculated between the prediction results of each model and the results of the other model. Hyperparameter settings are the same as those used in other methods.

4) The fourth group consists of traditional SSLDL algorithms, referred to as SA-LDL [18]. Since SA-LDL is an SSLDL algorithm designed for tabular data, we needed to perform feature engineering on image data, first, we use ResNet-50 for feature extraction from all datasets, followed by dimensionality reduction to 128 dimensions using PCA. For the remaining settings, we adhere to the defaults as specified in the paper.

5) The finally category consists of existing LDL algorithms. As there is currently only one open-source SSLDL algorithm, which is SA-LDL [18], we compared it with some state-of-the-art LDL algorithms. In this regard, we selected four state-of-the-art LDL algorithms: Adam-LDL-SCL [20], sLDLF [50], DF-LDL [13], and LDL-LRR [21]. These algorithm settings are defaulted to be consistent with those specified in the paper. Additionally, for these algorithms, we directly use labeled data to train the classifier. Then, we use the trained model to assign pseudo-labels to the unlabeled samples. Finally, we use the pseudo-labels to update the model.

**Evaluation Metrics**: We evaluate LDL algorithms using six metrics: five distance-based (Chebyshev, Clark, Kullback-Leibler, and Canberra) and two similarity-based (Cosine and Intersection). Lower values indicate better performance for distance-based metrics ($\downarrow$), while higher values indicate better performance for similarity-based metrics ($\uparrow$).

Table 5: The distribution distance/similarity measures and ranking correlation metrics

| Measure / Metric | Formula |
| --- | --- |
| Chebyshev $\downarrow$ | $\text{Dis}_1(\boldsymbol{d}, \hat{\boldsymbol{d}}) = \max_j \left| d_j - \hat{d}_j \right|$ |
| Clark $\downarrow$ | $\text{Dis}_2(\boldsymbol{d}, \hat{\boldsymbol{d}}) = \sqrt{\sum_{j=1}^{c} \frac{(d_j - \hat{d}_j)^2}{(d_j + \hat{d}_j)^2}}$ |
| Canberra $\downarrow$ | $\text{Dis}_3(\boldsymbol{d}, \hat{\boldsymbol{d}}) = \sum_{j=1}^{c} \frac{|d_j - \hat{d}_j|}{d_j + \hat{d}_j}$ |
| Kullback-Leibler $\downarrow$ | $\text{Dis}_4(\boldsymbol{d}, \hat{\boldsymbol{d}}) = \sum_{j=1}^{c} d_j \ln \frac{d_j}{\hat{d}_j}$ |
| Cosine $\uparrow$ | $\text{Sim}_1(\boldsymbol{d}, \hat{\boldsymbol{d}}) = \frac{\sum_{j=1}^{c} d_j \hat{d}_j}{\sqrt{\sum_{j=1}^{c} d_j^2} \sqrt{\sum_{j=1}^{c} \hat{d}_j^2}}$ |
| Intersection $\uparrow$ | $\text{Sim}_2 = \frac{1}{n} \sum_{i=1}^{n} \sum_{j=1}^{c} \min\left( d_{\boldsymbol{x}_i}^{y_j}, \hat{d}_{\boldsymbol{x}_i}^{y_j} \right)$ |
| Spearman's rank $\rho_S \uparrow$ | $\rho_S = 1 - \frac{6 \sum_i (\overline{D} - D)^2}{k(k^2 - 1)}$ |
| Kendall tau correlation $\tau_K \uparrow$ | $\tau_K = \frac{n_c - n_d}{\frac{1}{2}k(k-1)}$ |

## C.1 The Rest Experimental Results

**Convergence Analysis** Fig. 6 illustrates the convergence curves of the RankMatch algorithm on the Flickr-LDL and Twitter-LDL datasets. The rapid decline in the initial loss for Flickr-LDL indicates quick adaptation and efficient optimization during early epochs, stabilizing as the model converges. On the other hand, Twitter-LDL demonstrates a more gradual decline, reflecting a steadier learning process. These results confirm the robust optimization capability of RankMatch across diverse datasets.

## C.2 Parameter Sensitivity Analysis

To investigate the robustness of our method with respect to the trade-off parameter $\lambda$ in the PRR regularization term, we conduct a sensitivity analysis on four datasets: Emotion6, Flickr-LDL, RAF-LDL, and Twitter-LDL. As shown in Fig. 7, the performance remains stable across a wide range of $\lambda$ values from 0.005 to 0.1, demonstrating that the proposed framework is not overly sensitive to this hyperparameter. A small $\lambda$ (e.g., 0.01) generally achieves the best or near-best results across most metrics, indicating that a moderate contribution from the PRR loss is sufficient to capture label-ranking consistency without dominating the primary KL-divergence objective. These results validate the robustness and general applicability of the proposed PRR-regularized SSLDL framework across diverse datasets.

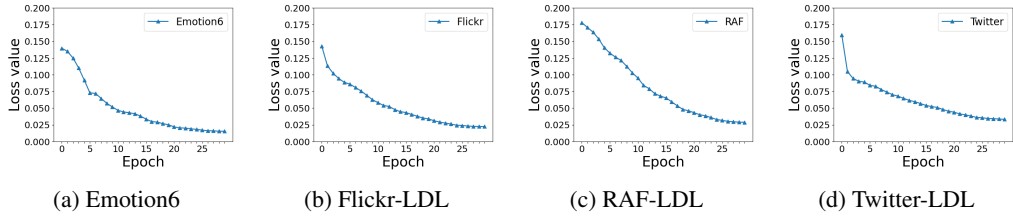

(a) Emotion6  (b) Flickr-LDL  (c) RAF-LDL  (d) Twitter-LDL

Figure 6: The convergence curve on Emotion6, FLickr-LDL, RAF-ML, and Twitter-LDL.

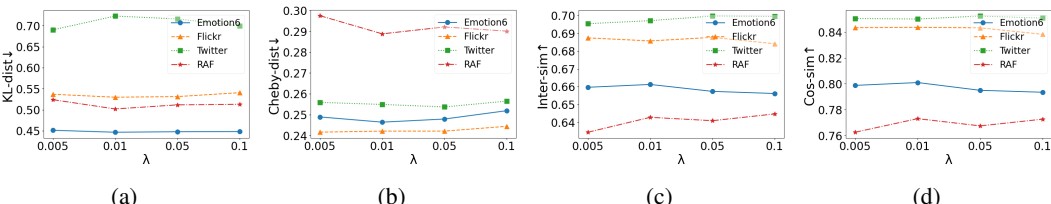

(a)  (b)  (c)  (d)

Figure 7: Parameter Sensitivity Analysis on 4 datasets.

