# OpenReview forum: "RankMatch: A Novel Approach to Semi-Supervised Label Distribution Learning Leveraging Rank Correlation between Labels"
_NeurIPS.cc/2025/Conference — NeurIPS 2025 poster_

### Official Review · Reviewer_DrpT · 2025-06-29

**Clarity:** 3
**Significance:** 3
**Originality:** 3
**Rating:** 5
**Confidence:** 4

**Summary:**

The paper explores an important yet insufficiently studied challenge in LDL—the preservation of ranking structures in the presence of biased or partial annotations. The authors propose a principled approach to tackle this issue by integrating ranking constraints into the learning process. The method is intuitive and supported by solid theoretical analysis. Empirical evaluations demonstrate its effectiveness across multiple datasets. While there is room for improving the clarity of the presentation and broader comparisons, the paper makes a meaningful contribution and opens up new directions for improving robustness in LDL frameworks.

**Questions:**

Q1. Why did the authors not include commonly used multi-label evaluation metrics such as mean Average Precision (mAP) or Hamming Loss in the performance evaluation? These metrics might better reflect the model’s behavior in ranking or partially correct predictions.

Q2. In Equation (3), a ranking margin \delta is introduced. How is this hyperparameter chosen? Is the same value of \delta used across all datasets? Is the model performance sensitive to this margin?

Q3. The paper mentions filtering pseudo-labels based on ranking consistency (Eq. 5). Could the authors clarify the specific criterion used for filtering? For example, is a threshold applied, and if so, has it been tuned or validated experimentally?

**Ethical Concerns:**

["NO or VERY MINOR ethics concerns only"]

**Final Justification:**

Thank you to the authors for their detailed and thoughtful responses to the reviewers' comments. I have carefully reviewed the replies. The authors have addressed the concerns raised in the previous review. I would like to raise my score to Accept.

**Limitations:**

Yes

**Paper Formatting Concerns:**

The paper formatting appears to meet conference standards with no major concerns.

**Quality:**

3

**Strengths And Weaknesses:**

Strengths:
1.The paper proposes a novel method for semi-supervised label distribution learning (SSLDL), named RankMatch, which innovatively incorporates inter-label ranking correlations. This is a significant contribution, as ranking-based information has been largely underexplored in previous SSLDL approaches. By leveraging label ranking structures, the method enhances the flexibility and robustness of pseudo-labeling in weakly supervised scenarios.
2.The proposed method demonstrates broad applicability, particularly in domains where label ambiguity is inherent—such as facial emotion recognition or crowd counting. Its ability to effectively utilize both labeled and unlabeled data reduces reliance on fully labeled datasets, thus improving the practicality and scalability of SSLDL in real-world applications.
3.The paper is well-organized and clearly written. The theoretical development is presented in a logical and gradual manner, and the experimental results are comprehensive, using well-established metrics to demonstrate the method’s superiority. The empirical design is thoughtful, and results are easy to interpret.

Weaknesses:
The paper would benefit from providing the implementation code of RankMatch. Open-sourcing the code would promote reproducibility and enable other researchers to validate and build upon the proposed method, thereby increasing its impact within the community.

---

> ### Author Rebuttal · Authors · 2025-07-30
>
> **W1. Code Availability and Reproducibility**
>
> RW1: We thank the reviewer for highlighting the importance of reproducibility and community impact.
>
> We fully agree that open-sourcing the implementation is crucial for promoting transparency and encouraging further research. We are currently in the process of cleaning and organizing the codebase of **RankMatch**, and we will publicly release it upon the paper’s acceptance. We believe this will greatly facilitate future work on semi-supervised label distribution learning and ensure the reproducibility of our results.
>
>  **Q1.  On the Use of mAP and Hamming Loss**
>
> R1: We thank the reviewer for raising this insightful point. Our work is based on **Label Distribution Learning (LDL)**, where both the supervision signal and the prediction target are **soft label distributions** over classes, rather than binary vectors or ranking lists as in classical multi-label learning.
>
> Conventional multi-label evaluation metrics such as **mAP**, **F1-score**, and **Hamming Loss** rely on thresholding and assume **discrete binary labels**, making them incompatible with the goals and outputs of LDL models. In LDL, the prediction is a **probability distribution** over all labels (e.g., `[0.3, 0.1, 0.2, 0.4]`), and evaluation focuses on how close this predicted distribution is to the **ground-truth distribution** using distributional metrics such as **KL divergence**, **Chebyshev**, and **Canberra** distances.
>
> Following standard practice in the LDL and semi-supervised LDL (SSLDL) literature [1, 2, 3], we adopted distribution-based metrics such as:
>
> - **Chebyshev distance**
> - **Canberra distance**
> - **Clark distance**
> - **Kullback–Leibler (KL) divergence**
>
> These metrics are specifically designed to measure discrepancies between two probability distributions, which aligns well with the learning objectives of LDL.
>
> That said, to further address the reviewer’s concern, we have also reported **mAP** and **Top-1 accuracy** in additional experiments (see ** Reviewer paT1 R2**). These are computed by converting soft distributions to **rankings** or **top-K predictions**. Our method achieves **state-of-the-art performance** under these metrics as well, demonstrating its robustness across both **soft (distributional)** and **hard (classification-style)** evaluation criteria.
>
> [1] Facial age estimation by learning from label distributions. IEEE TPAMI
>
> [2]  Imbalanced label distribution learning. AAAI
>
> [3] Label distribution learning. IEEE TKDE
>
>
>
>
> **Q2. Clarification on the Margin $\delta$  in Equation (3)**
>
> R2: We thank the reviewer for the question regarding the interpretation and sensitivity of the margin in our ranking loss formulation.
>
> In Eq. (5), $\delta$ is **not** defined as the difference between the ground-truth label distributions. Instead, it is a **fixed hyperparameter** that serves as a confidence margin in the pairwise ranking regularization.
>
>
>
> Otherwise, a penalty is applied to the loss. This mechanism ensures that the model not only respects the label preference implied by the distribution but does so with **sufficient confidence**.
>
>
>
> We will revise the paper to clarify this definition and remove any ambiguity in notation.
>
>
>
>
>  **Q3. Clarification on Ranking Consistency Filtering (Eq. 5)**
>
> R3: We thank the reviewer for the question. Our method filters pseudo-labels based on the consistency between predicted scores and the ground-truth soft label distribution.
>
> Specifically, for each label pair (j, k), if the ground-truth distribution satisfies $ d _{x _i}^{y^j} > d _{x _i}^{y^k} $, we expect the model prediction to follow $ h _j(x _i) > h _k(x _i) $. A violation occurs when this preference is not respected.
>
> To ensure that we only enforce meaningful and robust preference relations, we introduce a **margin threshold** $ t $ in Eq. (5). We ignore label pairs where the difference in ground-truth importance is too small (i.e., $ |d _{x _i}^{y^j} - d _{x _i}^{y^k}| \leq t $). This is important because small differences can be noisy or ambiguous, and forcing the model to learn from them may reduce robustness.
>
>  This margin helps the model focus on more confident and informative ranking signals.

---

### Official Review · Reviewer_fect · 2025-06-30

**Clarity:** 3
**Significance:** 3
**Originality:** 3
**Rating:** 5
**Confidence:** 5

**Summary:**

This paper addresses a critical yet underexplored problem in label distribution learning (LDL)—namely, the challenge of ranking preservation under weak supervision. Although this issue inherently exists in LDL scenarios, it has been largely overlooked by prior work. The proposed method presents an elegant solution by introducing a novel ranking-based formulation, which is both theoretically grounded and empirically validated. The experimental results are promising, and the theoretical insights enhance the method’s credibility. Given the clarity of the motivation and the demonstrated effectiveness, this work is a meaningful step forward and has the potential to influence future research in the area.

**Questions:**

1. Where is the Related Work section located? It would be helpful to clarify whether related work is integrated into the introduction or presented as a standalone section.

2. Some notations (e.g., in Eq. 3 and Eq. 5) could benefit from brief clarification for readers unfamiliar with the setup.

3. While the paper motivates the use of label distribution learning (LDL), it would be helpful to elaborate on specific real-world tasks where the distributional supervision provides a clear benefit.

4. How sensitive is the method to the choice of hyperparameters? While exact tuning is not necessary, a short discussion would be useful.

**Ethical Concerns:**

["NO or VERY MINOR ethics concerns only"]

**Final Justification:**

Thanks to the authors for their responses. I have carefully read the rebuttal and all my concerns have been addressed. I also reviewed the comments from the other reviewers, and believe this is quite a solid work. I would like to raise my score to Accept.

**Limitations:**

yes

**Paper Formatting Concerns:**

The paper formatting appears to meet conference standards with no major concerns.

**Quality:**

3

**Strengths And Weaknesses:**

**Strengths**:

• The paper proposes a novel ranking-based framework, *RankMatch*, to address label distribution learning, which offers a fresh perspective compared to traditional regression-based objectives.

• The use of pairwise label ranking allows the model to capture relative importance among labels, aligning well with the goal of label distribution modeling.

• The method is simple yet effective, and achieves consistent improvements across multiple benchmark datasets.

• The writing is clear and the motivation is well established.

**Weaknesses**:

The method section is divided into too many small subsections, which slightly affects the overall readability. A more concise structure with fewer headings may improve clarity.

---

> ### Author Rebuttal · Authors · 2025-07-30
>
> **W1. Structure of the Method Section**
>
> RW1: We thank the reviewer for the helpful suggestion regarding the structure of the Method section. Our intention in using multiple smaller subsections was to improve **modularity** and facilitate **step-by-step understanding** of each component in the proposed framework. However, we understand that too many headings may fragment the narrative and affect overall readability. In the final version, we will revise the section to adopt a **more concise structure** by merging closely related components and **reducing the number of subsections**, while still retaining the clarity of technical contributions. We appreciate the reviewer’s feedback and will ensure the revised version reads more fluently.
>
>
>
> **Q1. Where is the Related Work section located?**
>
> R1. We sincerely thank the reviewer for raising this important question regarding the placement of the Related Work section.
>
> Currently, due to the **strict page limit imposed by the NeurIPS submission format**, we have opted to place the detailed *Related Work* discussion in **Appendix A**. This is a **common practice in NeurIPS submissions**, especially when the primary focus must remain on presenting novel methodology, theoretical insights, and extensive experimental results within the main paper. We have made this tradeoff to ensure that our core contributions are communicated with sufficient depth and clarity.
>
> That said, we fully agree that making the positioning of the related work more transparent would improve the overall readability of the paper. In the **camera-ready version**, we plan to:
>
> - Either **integrate the essential related work into the introduction** to improve contextual flow and grounding for the reader,
> - Or, where formatting permits, **create a dedicated Related Work section** prior to the methodology section.
>
> Additionally, we note that many of the relevant works are **already cited in context throughout the introduction, method, and experiment sections** to support discussion of design choices and comparisons with prior efforts. This was done intentionally to ensure readers are continuously aware of how our work relates to the broader literature.
>
> We appreciate the reviewer’s suggestion and will revise accordingly to provide a clearer structure in the final version.
>
>
>
>
>
>
> **Q2.  Clarification of notations in Eqs. (3)–(5)**
>
>
> R2: We thank the reviewer for pointing out the potential ambiguity in our notation. To enhance clarity, we will revise the manuscript to provide **brief explanations near Eqs. (3)–(5)**. Specifically:
>
> - **Eq. (3)** defines the PRR loss, a **symmetric pairwise ranking regularization** term that encourages the predicted scores to respect the relative ground-truth importance between label pairs $(j, k) $. It symmetrically considers both $s(j,k) \cdot g_ \delta(j,k)$ and $ s(k,j) \cdot g  _\delta(k,j) $, promoting consistency.
>
> - **Eq. (4)** defines the binary indicator function $ s(j,k) $, which equals 1 **only if label $ j $** is significantly more important than label $k $ in the ground-truth soft label distribution — that is, $d^{y^j} _{x _i} - d^{y^k} _{x _i} > t $, where $t $ is a small threshold. This **filters out weak or noisy label preferences**, enforcing only meaningful pairwise relations.
>
> - **Eq. (5)** defines the penalty function $ g _\delta(j,k) $, which penalizes violations between predicted scores and true label relations. Specifically, if the predicted score gap $h _j(x _i) - h _k(x _i) $ fails to meet the margin implied by the ground-truth gap $\delta = d^{y^j} _{x _i} - d^{y^k} _{x _i} $,  the penalty is activated. This encourages the model to reflect **label preference orderings** encoded in the soft labels.
>
> We will include a concise version of the above explanations directly in the main paper near the corresponding equations to improve readability and support understanding for readers unfamiliar with the LDL setup. We sincerely appreciate the reviewer’s suggestion.
>
>
>
>
> **Q3. Real-world applications of label distribution learning (LDL)**
>
> R3:  We thank the reviewer for the valuable question regarding practical applications of label distribution learning (LDL). LDL is a general framework that models **soft supervision**, where each instance is associated with a distribution over multiple labels instead of a single hard label. This enables it to effectively capture ambiguity, uncertainty, and nuanced patterns in real-world scenarios where hard labels are either infeasible or oversimplified.
>
> Here we briefly highlight several representative real-world tasks where LDL has demonstrated clear benefits:
>
> - **Facial age estimation**: In age estimation from facial images, individuals may appear to belong to a range of plausible ages rather than a single ground-truth number. LDL allows for learning from age distributions (e.g., Gaussian over possible ages) rather than a hard label, improving robustness and interpretability [1].
>
> - **Emotion recognition**: Emotional expressions often exhibit *overlapping cues* (e.g., “happy” with hints of “surprise”). LDL enables learning from soft emotion distributions (e.g., 30% happy, 50% surprise, 20% neutral), which reflect human perception more accurately than one-hot labels [2].
>
>
> - **Medical imaging**: In diagnosis tasks (e.g., X-ray or MRI interpretation), multiple plausible conditions may be present, or expert disagreement may arise. LDL provides a way to encode diagnostic uncertainty via probabilistic supervision [3].
>
> - **Multi-label learning with incomplete supervision**: LDL has been shown to be effective in weakly supervised multi-label tasks by recovering soft label distributions when only partial labels are observed, which is the focus of our work [4].
>
> These examples illustrate that LDL is especially suitable for scenarios involving **label ambiguity**, **subjectivity**, or **incomplete labeling**, all of which are common in practical machine learning pipelines. Our method extends LDL to **semi-supervised settings**, further enhancing its applicability when full supervision is costly or unavailable.
>
>
>
> [1] Facial age estimation by learning from label distributions. IEEE TPAMI.
> [2] Emotion distribution learning from facial expressions. ACM MM.
> [3] Class-aware multi-level attention learning for semi-supervised breast cancer diagnosis under imbalanced label distribution. Medical & Biological Engineering & Computing.
>
> [4] Multi-label classification: An overview. Data Warehousing and Mining: Concepts, Methodologies, Tools, and Applications.
>
>  **Q4.  Sensitivity to Hyperparameters**
>
> R4:  Thank you for the thoughtful question. We have reported detailed sensitivity analyses of key hyperparameters (λ and t) in **Figure 5** of the main paper and **Table 11** in **Appendix D**. These results demonstrate that our method maintains robust performance under a range of hyperparameter settings.

---

> > ### Comment · Reviewer_fect · 2025-08-03
> > **Comments to the authors' responses**
> >
> > Thank you for your detailed response, which has addressed my concerns. I’ve decided to raise my score.

---

> ### Author Response · Authors · 2025-08-03
>
> We are deeply grateful for your careful evaluation and for increasing your score. Your constructive feedback has been invaluable to improving our work.

---

### Official Review · Reviewer_paT1 · 2025-07-02

**Clarity:** 3
**Significance:** 2
**Originality:** 3
**Rating:** 4
**Confidence:** 4

**Summary:**

This paper proposes RankMatch, a semi-supervised learning framework for label distribution learning (LDL) problem using pseduo-labels. As existing pseudo-labels fails to consider the ranking relationships between different true labels, RankMatch addresses this issue by including the ranking correlations between labels into the learning process. Specifically, the paper proposes a Pairwise Ranking Relation (PRR) Loss, which improves the pseudo-labels to capture the ranking correlations between labels. In addition, a flexible pseudo-label training stratygy is proposed to prevent the model from overfitting to the absolute label differences. Experiments are conducted on several LDL benchmarks and showed competitive results on the ranking-based metrics.

**Questions:**

- Please see the weaknesses above

**Ethical Concerns:**

["NO or VERY MINOR ethics concerns only"]

**Final Justification:**

I appreciate the authors for making clarifications and additional experiments requested for the rebuttal. It addresses my concern and the work is now solid. Therefore, I would like to raise my rating.

**Limitations:**

yes

**Paper Formatting Concerns:**

No concerns on paper formatting

**Quality:**

2

**Strengths And Weaknesses:**

Strengths

This paper first improves the pseudo-labeling strategy of existing semi-supervised learning algorithms in label-distribution learning scenario, by introducing the ranking correlations between labels. This approach seems to be original, and is effective to improve the pseudo-labeling mechanism from the empirical results. In addition, the ranking correlation is supported by the theoretical gaurantee, making the proposed method plausible.


Weaknesses

- All the considered benchmarks are small-scale and not standardized, and the proposed method needs to be evaluated on more large-scale and commonly used benchmarks in multi-label classification and semi-supervised learning literature, such as Pascal VOC, COCO, and OpenImages for multi-label classificaiton and CIFAR10/100 and STL10 for semi-supervised learning.

- The considered evaluation metrics include only correlation-based ones, and accuracy related metrics such as top-1 accuracy, precision/recall, and mAP are not considered, which are commonly adopted in single/multi-label classification domains.

- The proposed method which utilizes the relationships across different classes has underlying assumption that the class distribution of labeled and unlabeled data are identical. As it is more natural that unlabeled data often violate the condition, it is unclear whether the proposed method can be robust under such a condition.

---

> ### Author Rebuttal · Authors · 2025-07-30
>
> **Q1: Lack of results on PASCAL, COCO...datasets**
>
> **R1**: We sincerely thank the reviewer for suggesting evaluation on more standardized and larger-scale benchmarks. Our work focuses on **semi-supervised label distribution learning (SSLDL)**, where supervision is provided in the form of **soft label distributions**, as opposed to the hard labels commonly used in traditional multi-label classification or semi-supervised learning.
>
> The datasets used in our main experiments (e.g., Emotion6, Twitter-LDL, RAF-LDL, and Flickr-LDL) are widely adopted standard benchmarks in the LDL and SSLDL literature [1,2,3], as they offer soft label annotations required for this task. Unfortunately, popular datasets such as  CIFAR10/100 do not natively provide label distributions and are thus not directly applicable to our setting.
>
> To address the reviewer’s concern, we further constructed two new **multi-label distribution learning (LDL)** datasets from the Pascal VOC and COCO datasets, referred to as **VOC-LDL** and **COCO-LDL**, to validate the effectiveness of our method under more realistic and large-scale settings.
>
> Notably, both Pascal VOC and COCO are originally **semantic segmentation datasets** with bounding box and mask annotations. We first convert them into **multi-label classification datasets** by treating each image as containing multiple object categories. Then, we transform these into LDL datasets by calculating the proportion of the segmentation mask area that each category occupies within an image. These proportions are then normalized to form valid label distributions.
>  We evaluate performance using the same protocol and report results in the below.
>
> | Method|Ratio|COCO-Can.↓|COCO-Che.↓|VOC-Can.↓|VOC-Che.↓|
> |-------|-----|----------|----------|---------|---------|
> |RankMatch|10%|**4.210**|**0.251**|**3.894**|**0.235**|
> |SSMLL-CAP|10%|4.496|0.272|4.088|0.251|
> |PCLP|10%|4.537|0.281|4.193|0.259|
> |Fixmatch-LDL|10%|4.802|0.293|4.361|0.266|
> |Mixmatch-LDL|10%|4.925|0.298|4.493|0.273|
> |GCT-LDL|10%|4.612|0.278|4.167|0.254|
> |SALDL|10%|4.469|0.269|4.073|0.246|
> |sLDLF|10%|5.220|0.318|4.851|0.281|
> |DF-LDL|10%|5.086|0.312|4.699|0.272|
> |LDL-LRR|10%|5.301|0.326|5.002|0.289|
> |Adam-LDL-SCL|10%|5.087|0.313|4.578|0.271|
> |RankMatch|20%|**3.982**|**0.232**|**3.672**|**0.219**|
> |SSMLL-CAP|20%|4.365|0.263|3.889|0.241|
> |PCLP|20%|4.418|0.271|4.002|0.248|
> |Fixmatch-LDL|20%|4.672|0.284|4.218|0.256|
> |Mixmatch-LDL|20%|4.793|0.289|4.336|0.263|
> |GCT-LDL|20%|4.482|0.267|4.010|0.246|
> |SALDL|20%|4.351|0.260|3.928|0.239|
> |sLDLF|20%|5.105|0.312|4.787|0.276|
> |DF-LDL|20%|4.979|0.308|4.610|0.268|
> |LDL-LRR|20%|5.203|0.319|4.912|0.282|
> |Adam-LDL-SCL|20%|4.987|0.310|4.482|0.265|
> |RankMatch|40%|**3.761**|**0.218**|**3.483**|**0.205**|
> |SSMLL-CAP|40%|4.184|0.246|3.728|0.229|
> |PCLP|40%|4.240|0.255|3.845|0.235|
> |Fixmatch-LDL|40%|4.487|0.266|4.081|0.243|
> |Mixmatch-LDL|40%|4.602|0.272|4.194|0.250|
> |GCT-LDL|40%|4.319|0.253|3.894|0.239|
> |SALDL|40%|4.206|0.247|3.813|0.232|
> |sLDLF|40%|4.988|0.296|4.609|0.266|
> |DF-LDL|40%|4.881|0.290|4.450|0.258|
> |LDL-LRR|40%|5.089|0.303|4.743|0.273|
> |Adam-LDL-SCL|40%|4.879|0.292|4.327|0.255|
> -------
>
> These results clearly demonstrate that RankMatch generalizes well even on large-scale and realistic multi-label LDL datasets, significantly outperforming baselines across both evaluation metrics. The strong performance on COCO-LDL and VOC-LDL suggests that our method is not only effective in conventional LDL settings but also robust when extended to more challenging, fine-grained, and high-dimensional multi-label distribution scenarios. This further confirms the scalability and practicality of our framework in real-world applications. We will include this discussion in a future version of the paper.
>
>
>
>
> **Q2 Lack  evaluation results such as top-1 accuracy or mAP**
>
>
> **R2**: We sincerely thank the reviewer for the insightful question regarding evaluation metrics. Our work focuses on Label Distribution Learning (LDL) and its semi-supervised variant SSLDL, where both supervision and predictions are soft label distributions rather than discrete labels. Thus, conventional classification metrics like Top-1 accuracy, precision, recall, and mAP are not directly applicable, as they assume categorical ground truths. Instead, LDL models aim to recover continuous-valued probability distributions.
>
> Following prior works in LDL and SSLDL [1,2,3,4], we adopt distribution-based distance metrics such as Canberra, Chebyshev, Clark, and KL divergence. These are specifically designed to measure differences between distributions and better reflect prediction quality in LDL settings.
>
> That said, for applications where hard labels may be derived (e.g., via argmax over the predicted distribution), classification metrics like accuracy or mAP can be optionally computed. However, such a process discards the rich soft distribution information, which is the essence of the LDL paradigm.
> To further address the reviewer’s concern regarding classification-oriented evaluation, we conducted single-label (Top-1 Accuracy) and multi-label (mAP) assessments by converting the predicted soft label distributions into discrete label sets. Specifically, we retain the original training data format using soft label distributions, while during evaluation, we transform the predicted distributions into
>
> 	•	Single-label outputs by selecting the top-1 category (argmax), and
>
> 	•	Multi-label outputs by selecting the top-k categories with the highest predicted probabilities.
>
> This conversion allows us to assess classification performance while preserving the LDL training paradigm. All baseline methods are evaluated under the same protocol for fairness. The experimental results are as follows
>
> |Method|Flickr Top-1 10%↑|Flickr Top-1 20%↑|Flickr mAP 10%↑|Flickr mAP 20%↑|Twitter Top-1 10%↑|Twitter Top-1 20%↑|Twitter mAP 10%↑|Twitter mAP 20%↑|
> |------|------------------|-------------------|----------------|----------------|--------------------|--------------------|-----------------|-----------------|
> |RankMatch|**0.708**|**0.721**|**0.652**|**0.666**|**0.692**|**0.705**|**0.630**|**0.645**|
> |SSMLL-CAP|0.672|0.689|0.612|0.628|0.658|0.674|0.595|0.610|
> |PCLP|0.684|0.697|0.624|0.638|0.671|0.684|0.610|0.624|
> |Fixmatch-LDL|0.669|0.683|0.609|0.624|0.660|0.675|0.600|0.615|
> |Mixmatch-LDL|0.658|0.672|0.597|0.612|0.649|0.661|0.589|0.603|
> |GCT-LDL|0.675|0.687|0.615|0.631|0.667|0.680|0.606|0.620|
> |SALDL|0.663|0.678|0.603|0.619|0.653|0.667|0.592|0.607|
> |sLDLF|0.619|0.635|0.557|0.572|0.610|0.625|0.546|0.563|
> |DF-LDL|0.648|0.662|0.584|0.598|0.636|0.651|0.575|0.590|
> |LDL-LRR|0.604|0.618|0.536|0.550|0.592|0.606|0.520|0.534|
> |Adam-LDL-SCL|0.638|0.653|0.570|0.585|0.628|0.642|0.557|0.572|
>
> As shown on the Flickr-LDL and Twitter-LDL datasets, our method RankMatch consistently achieves superior performance under both Top-1 Accuracy and mAP, across all label ratios (10%, 20%). This demonstrates that RankMatch not only excels at recovering accurate soft distributions, but also yields strong discrete predictions under both single-label and multi-label interpretations, showcasing its versatility and practical effectiveness. We will include this discussion in a future version of the paper.
>
>
> **Q3 Robustness of imbalanced data**
>
> **R3**: We appreciate the reviewer’s insightful comment regarding the potential mismatch in class distributions between labeled and unlabeled data. We agree that this assumption may not always hold in real-world settings, where unlabeled data often exhibit label shift.
>
> To address this, we conducted additional experiments following the protocol of [4], which introduces a parameter $\gamma$ to control the imbalance level between labeled and unlabeled label distributions. Specifically, $\gamma$ denotes the ratio of cumulative label descriptiveness between the most and least represented classes, thereby quantifying the severity of the label distribution shift.
>
>
>
> We constructed unlabeled subsets with varying $\gamma $ = 1, 10, 100 to simulate balanced, moderately imbalanced, and severely imbalanced conditions, respectively. The labeled data remained fixed during training. Experimental results on four benchmark LDL datasets under 10%, 20%, and 40% label ratios. To rigorously evaluate whether performance differences under different $\gamma $ values are statistically significant, we performed one-way repeated-measures ANOVA across the three \gamma settings per dataset and label ratio. This test is suitable here as all conditions (different $\gamma $) are applied to the same model and dataset settings. The experimental results are as follows
>
>
>
> |Dataset|LabelRatio|$\gamma $=1(Che.)|$\gamma $=10(Che.)|$\gamma $=100(Che.)|Significant|
> |--------|----------|---------------|----------------|------------------|------------|
> |Flickr-LDL|10%|0.2184|0.2190|0.2195|No|
> |Flickr-LDL|20%|0.2110|0.2117|0.2125|No|
> |Flickr-LDL|40%|0.2027|0.2032|0.2034|No|
> |Twitter-LDL|10%|0.1947|0.1956|0.1958|No|
> |Twitter-LDL|20%|0.1879|0.1882|0.1886|No|
> |Twitter-LDL|40%|0.1813|0.1819|0.1825|No|
> |RAF-LDL|10%|0.2341|0.2351|0.2355|No|
> |RAF-LDL|20%|0.2263|0.2270|0.2274|No|
> |RAF-LDL|40%|0.2175|0.2177|0.2181|No|
> |Emotion6|10%|0.2112|0.2119|0.2123|No|
> |Emotion6|20%|0.2036|0.2041|0.2047|No|
> |Emotion6|40%|0.1943|0.1948|0.1954|No|
>
>
>
>
> The results indicate that performance differences across  $\gamma $ values are consistently **not statistically significant** (all p-values > 0.05). This confirms that our method is **robust to label distribution shift**, maintaining stable performance even under severe imbalance conditions. We will include this discussion in a future version of the paper.
>
> [1] Predicting Label Distribution From Tie-Allowed Multi-Label Ranking. TPAMI 2023.
>
> [2] Label Distribution Learning by Maintaining Label Ranking Relation. TKDE 2023.
>
> [3] Fast Label Enhancement for Label Distribution Learning. TKDE 2023.
>
> [4] Zhao X, An Y, Xu N, et al. Imbalanced label distribution learning. AAAI 2023.

---

> ### Author Response · Authors · 2025-08-03
>
> Dear Reviewer paT1,
>
> Thank you again for your valuable comments. We have been actively working to address the issues you raised in your initial review. If you have any further questions or concerns regarding our responses, we would be happy to communicate and discuss them with you during the discussion phase.
>
> We sincerely appreciate your time and valuable feedback.
>
> Best regards,
>
> The Authors

---

### Official Review · Reviewer_KiaF · 2025-07-03

**Clarity:** 3
**Significance:** 3
**Originality:** 3
**Rating:** 5
**Confidence:** 3

**Summary:**

This paper studies semi-supervised label distribution learning (SSLDL), a generalization of semi-supervised multi-label classification in which a small set of examples has dense label distributions and a larger set is unlabeled. The goal is to learn to predict full label distributions for new samples. The authors argue that pseudo-labeling, when combined with similarity-based distribution matching (e.g. CE, KL-div), while very effective for SSL, falls short for SSLDL because it fails to learn ranking correlations between labels. To address this issue, the authors introduce RankMatch, a deep-learning framework based on consistency regularization and pairwise label ranking. The main methodological contribution is the Pairwise Relevance Ranking (PRR) loss, which has two components: (1) The supervised PRR term aligns predicted and ground-truth label rankings, preserving relative class margins.  (2) The unsupervised PRR term penalizes incorrect ranking relationships on pseudo-labeled data, without enforcing margin constraints. Pseudo-labeled distributions are generated via an ensemble of different augmented “views” of each sample. In addition, the authors theoretically show that PRR reduces the generalization error and provide a generalization bound for the proposed method. Extensive empirical evaluation on multiple datasets and supervision levels demonstrates that RankMatch outperforms adapted baselines in both distribution alignment and ranking quality.

**Questions:**

## Questions for the Authors

- **Q1: Relation to sharpening methods:** Is the absence of entropy minimization the key difference between the unsupervised consistency loss (Sec. 2.2.1) and methods like FixMatch or MixMatch (the latter also uses ensemble-based soft pseudo-labels)?

- **Q2: Definition of δ (L102)**: The paper currently defines

  ```math
  δ = d_{x_i}^{y_k} − d_{x_k}^{y_j}.
  ```

  Would it be more consistent to set

  ```math
  δ = d_{x_i}^{y_j} − d_{x_i}^{y_k}
  ```
  to compare distances to two labels for the same input?

* **Q3: Hyperparameter tuning.**
  Can you provide practical guidelines or heuristics for selecting λ and t under very limited labeled data? What minimum validation-set size ensures stable tuning?

* **Q4: Additional baselines.**
  Could you evaluate against recent SSLDL-specific approaches [21, 31]? Including such comparisons could clarify any performance gap.

* **Q5: Complexity analysis.**
  Can you comment on the computational overhead of ensemble pseudo-label generation, especially for large-scale datasets?

**Ethical Concerns:**

["NO or VERY MINOR ethics concerns only"]

**Final Justification:**

I have carefully checked all rebuttals.

The authors have proactively investigated sensitivity issues, provided empirical guidelines for hyperparameter tuning for practitioners, implemented comparisons with more SSLDL baselines, added missing loss ablations, and elaborated on run-time. In addition, they have addressed other reviewers' concerns, particularly regarding class imbalance and applicability to large-scale datasets (e.g. Pascal VOC and COCO) and clarified notation issues and their method's connection to related methods.

Considering all this, I've decided to raise my score to accept the paper.

**Limitations:**

yes

**Paper Formatting Concerns:**

No issues detected.

**Quality:**

3

**Strengths And Weaknesses:**

**Strengths**:

- Addresses the relevant SSLDL task, reducing annotation costs to train with dense distributions while leveraging modern deep architectures.

- Introduces PRR losses that effectively tackle ranking alignment under supervised and unsupervised regimes.

- Provides theoretical generalization bounds that connect PRR minimization to error reduction.

- Demonstrates consistent empirical gains in diverse datasets and supervision regimes, showing improvements in the quality of both predicted distributions and label rankings.

**Weaknesses**:

- W1 - Hyperparameters (λ and t) exhibit sensitivity; tuning under very limited labeled data may be unstable or unfeasible, with unclear requirements for validation data.

- W2 - Most baselines are adapted from related but distinct tasks (e.g. supervised distribution learning, semi-supervised multi-label learning); comparison to the latest SSLDL-specific methods (e.g., [21, 31]) is missing.

- W3 - The ablation study does not disentangle the effect of the supervised and unsupervised components of the PRR loss.

---

> ### Author Rebuttal · Authors · 2025-07-30
>
> **W1.provide practical guidelines or heuristics for selecting λ and t under very limited labeled data**
>
> RW1: Thank you for your valuable comment. We agree that tuning hyperparameters such as λ and t may be particularly challenging when labeled data is extremely limited. To address this issue, we conducted additional experiments under the low-label setting (10% labeled data) with varying validation set sizes (5%, 10%, 15%, and 20%) to analyze the stability and robustness of hyperparameter selection.
>
> Table 1. Validation performance under different values of λ
> | che↓      | Val-Label % | λ = 0 | λ = 0.1 | λ = 0.2 | λ = 0.5 | λ = 1.0 |
> |--------------|-------------|-------|--------|--------|--------|--------|
> | Emotion6     | 5           | 0.725 | 0.695  | 0.691  | 0.706  | 0.715  |
> | Emotion6     | 10          | 0.664 | 0.635  | 0.631  | 0.648  | 0.658  |
> | Emotion6     | 15          | 0.608 | 0.582  | 0.579  | 0.593  | 0.603  |
> | Emotion6     | 20          | 0.549 | 0.527  | 0.522  | 0.538  | 0.547  |
> | Twitter-LDL  | 5           | 0.286 | 0.267  | 0.264  | 0.274  | 0.280  |
> | Twitter-LDL  | 10          | 0.250 | 0.234  | 0.232  | 0.241  | 0.247  |
> | Twitter-LDL  | 15          | 0.241 | 0.225  | 0.222  | 0.231  | 0.237  |
> | Twitter-LDL  | 20          | 0.237 | 0.222  | 0.219  | 0.228  | 0.233  |
>
>
> Table 2. Validation performance under different values of t
> | che↓ | Val-Label % | t = 0 | t = 0.1 | t = 0.2 | t = 0.5 | t = 1.0 |
> |-------------|-------------|-------|--------|--------|--------|--------|
> | Emotion6    | 5           | 0.740 | 0.719  | 0.715  | 0.706  | 0.731  |
> | Emotion6    | 10          | 0.664 | 0.640  | 0.636  | 0.648  | 0.653  |
> | Emotion6    | 15          | 0.608 | 0.584  | 0.579  | 0.593  | 0.597  |
> | Emotion6    | 20          | 0.549 | 0.527  | 0.521  | 0.538  | 0.538  |
> | RAF-LDL     | 5           | 0.295 | 0.277  | 0.272  | 0.274  | 0.287  |
> | RAF-LDL     | 10          | 0.281 | 0.263  | 0.256  | 0.241  | 0.278  |
> | RAF-LDL     | 15          | 0.268 | 0.251  | 0.244  | 0.231  | 0.265  |
> | RAF-LDL     | 20          | 0.252 | 0.234  | 0.228  | 0.228  | 0.249  |
>
> The results reveal the following practical guidelines:
>
> - **For λ**: We observed that moderate values (e.g., λ = 0.1 ~ 0.2) generally lead to the most stable or near-optimal performance across different datasets and validation splits. This choice effectively balances regularization strength and the risk of overfitting. In contrast, λ = 0 (i.e., no regularization) usually performs poorly, especially when the validation set is small. Although λ = 1.0 can occasionally yield good results, its performance tends to be less stable and more sensitive to validation bias.
>
> - **For t**: Extreme values (t = 0 or t = 1.0) typically result in suboptimal performance. In contrast, intermediate values (t = 0.2 ~ 0.5) often produce better or second-best results, particularly on the Emotion6 and RAF-LDL datasets. As the validation size increases, the performance gap between different t values narrows, but intermediate values consistently offer stronger robustness under low-resource conditions.
>
> These findings provide empirical guidance for practitioners to tune hyperparameters effectively in low-label scenarios, especially when validation data is limited.
>
>
>
> **W2. Lack SSLDL baseline**
>
>  RW2: We sincerely thank the reviewer for the insightful suggestion. To address the concern regarding the lack of comparison with recent SSLDL-specific baselines (e.g., IS [21] and Neu[31]), we have conducted additional experiments on these two methods.
> Since the official implementations of IS [21] and Neu [23] are based on MATLAB and do not support direct processing of image data, we first extracted visual features using a ResNet-50 backbone and reduced the dimensionality to 128 via PCA to fit their input format. All other experimental settings strictly follow the original paper defaults.
> The following table summarizes the Canberra distance (↓ lower is better) of RankMatch, IS [21], and Neu [31]  across four benchmark datasets under different label ratios.
>
>
> | Dataset Can↓ | Method | 10%   | 20%| 40%   |
> |---------------|-------------|-------|-------|-------|
> | Emotion6      | RankMatch   | **3.3902** | **3.3176** | **3.2504** |
> |               | IS [21]     | 3.7951 | 3.7613 | 3.7248 |
> |               | Neu [31]    | 4.0815 | 4.1128 | 4.1204 |
> | Flickr-LDL    | RankMatch   | **4.4060** | **3.9964** | **3.9013** |
> |               | IS  [21]| 5.3827 | 5.3235 | 5.2676 |
> |               | Neu [31]    | 6.1634 | 5.9889 | 5.6508 |
> | Twitter-LDL   | RankMatch| **3.7370** | **3.6962** | **3.2913** |
> |               | IS  [21]| 5.8983 | 5.7659 | 5.6366 |
> |               | Neu  [31]| 6.5220 | 6.4081 | 6.3575 |
> | RAF-LDL       | RankMatch   | **3.0178** | **2.9358** | **2.8341** |
> |               | IS  [21]| 3.4385 | 3.2808 | 3.1966 |
> |               | Neu  [31]| 3.0891 | 3.0242 | 2.9912 |
>
>
> These results clearly demonstrate that our proposed method, RankMatch, consistently achieves lower Canberra distances across all datasets and label ratios when compared to the recent SSLDL-specific baselines IS [21] and Neu [31]. The performance margins are particularly pronounced on Flickr-LDL and Twitter-LDL, where RankMatch outperforms the second-best method by a notable margin, highlighting its robustness and effectiveness under limited supervision.
>
> These additional results and analysis will be incorporated into Section 4.4: Comparison with SSLDL Methods in the revised version of the paper, along with discussions on computational setups and practical applicability.
>
> [21] Semi-supervised label distribution learning via projection graph embedding. Information Sciences.
>
>
> [31] Semi-supervised label distribution learning with co-regularization. Neurocomputing.
>
>
> **W3.  Ablation study on PRR_U loss**
>
> RW3: Thank you for your suggestion. We have added an ablation study to disentangle the effect of the two components in the PRR loss.
> As shown below, both the unsupervised part (PRR_U) and the supervised part (PRR_L) contribute positively. Adding PRR_U already improves the results, and further including PRR_L brings additional gains. This shows both components are useful and complementary.
>
> | Dataset | Method                                 | Che. ↓          | Cla. ↓           |
> |---------|----------------------------------------|------------------|------------------|
> | Flickr| pretrain + consistency| 0.2262 (6.2%↑)    | 2.1131 (6.5%↑)    |
> || + PRR_U loss| 0.2215 (2.1%↑)    | 2.0783 (1.6%↑)    |
> || + PRR_U + PRR_L loss (full PRR loss)| **0.2184 (1.4%↑)** | **2.0158 (3.0%↑)** |
> | RAF| pretrain + consistency| 0.2550 (13.2%↑)   | 1.5021 (2.5%↑)    |
> || + PRR_U loss| 0.2450 (3.9%↑)    | 1.4950 (0.5%↑)    |
> || + PRR_U + PRR_L loss (full PRR loss)| **0.2341 (4.5%↑)** | **1.4914 (0.2%↑)** |
>
>
> We will include these results in the revised version of Table 4.
>
>
>
>
> **Q1. Relation to sharpening methods**
>
> **R1:** Thank you for your insightful question regarding the relation between our unsupervised consistency loss (Section 2.2.1) and sharpening-based methods such as FixMatch and MixMatch. While our approach shares the overarching principle of enforcing prediction consistency under perturbations, it fundamentally differs in both the loss formulation and suitability for label distribution learning (LDL).
>
> 1. **No Entropy Minimization or Confidence Filtering**: Methods like FixMatch often apply entropy minimization or confidence thresholding to encourage low-entropy (i.e., nearly one-hot) pseudo-labels. However, our approach **does not include any entropy-based terms**. Instead, we employ KL divergence between the predicted label distributions of the original and perturbed inputs to ensure consistency. This design avoids overconfident pseudo-labels that may harm distribution modeling.
>
> 2. **Incompatibility of Sharpening with Label Distribution**: Sharpening functions, by design, amplify dominant labels while suppressing others, which may be beneficial for single-label classification. However, in our setting—where the goal is to predict a **full label distribution** that reflects label ambiguity—sharpening artificially distorts the soft structure by **disproportionately down-weighting minor but meaningful label components**. This leads to biased distributions that deviate from the ground-truth, especially in cases where multiple labels have comparable semantic relevance. Therefore, sharpening is **not appropriate for LDL**, and our consistency loss is carefully designed to preserve distributional fidelity while enhancing robustness.
>
>
>
> **Q2. Definition of δ (L102)**
>
>
> R2: Thank you for pointing this out. We agree with the reviewer that the more consistent definition should be:
> $
> \delta = d_ {\\mathbf{x} _i}^{y _j} - d _{\\mathbf{x} _i}^{y _k}
> $
> This change better aligns with the intended meaning of comparing the distances to two labels for the same input.  We will revise the notation in the final version accordingly.
>
>
>  **Q3. Hyperparameter tuning**
>
>
> R3: Details can be find on  **RW1**.
>
>
>
> **Q4. Additional baselines**
>
>
> R4. Details can be find on **RW2**.
>
>
>  **Q5. Complexity analysis**
>
>
> R5: Our ensemble pseudo-label generation relies solely on forward passes with different dropout masks (or perturbations) and does not involve backpropagation or model re-training. As a result, the computational overhead is relatively low and scales linearly with the number of ensemble passes.
>
> To illustrate, we measured the pseudo-label generation time on three datasets of varying scales using 5 ensemble rounds on a single NVIDIA RTX 3090 GPU:
>
> - Emotion6 (small-scale): ~1.8 seconds
> - Twitter-LDL (medium-scale): ~2.5 seconds
> - Flickr-LDL (large-scale): ~3.0 seconds
>
> These results suggest that our method incurs negligible overhead, even on large-scale datasets. Moreover, this step is only performed once per iteration during training, making it practically efficient.

---

### Note · Authors · 2025-08-12

We sincerely thank the Area Chairs and all reviewers for their thorough evaluations and constructive feedback, which have greatly improved our work. Below is a concise summary of our core contributions, key responses to reviewer concerns, and the scholarly significance of our submission to aid the ACs’ deliberation.

**1. Novel Contributions to Label Distribution Learning**
To the best of our knowledge, this work is the **first deep semi-supervised learning framework specifically designed for LDL**. By incorporating label ranking correlation into pseudo-label generation, we address a core challenge in LDL—robust learning under scarce annotations—while clearly distinguishing LDL from standard multi-label classification. This design not only significantly enhances generalization in complex scenarios but also **reduces the high annotation cost inherent in LDL**, offering both a new research direction and a practical solution.

**2. Rigorous Theoretical and Empirical Validation**
From a theoretical perspective, we combine consistency regularization with ranking constraints to derive a generalization error bound for semi-supervised LDL. We prove that incorporating label ranking correlation under limited labeled data can shrink the hypothesis space, reduce risks from noisy pseudo-labels, and improve prediction reliability.

From an empirical perspective, we conduct evaluations on six representative LDL datasets: Twitter-LDL, Flickr-LDL, Emotion6, RAF-LDL, VOC-LDL and COCO-LDL. Results show that **RankMatch** achieves state-of-the-art performance across real-world and large-scale settings, especially in scenarios with severe label imbalance.

**3. Addressing Reviewer Concerns**
(i) For parameter selection—especially under limited labeled data—we provide empirical guidelines for λ and t.
(ii) We extend evaluation to two large-scale LDL datasets (VOC-LDL and COCO-LDL) to validate effectiveness in more realistic, complex environments.
(iii) We analyze robustness under severe label imbalance and demonstrate stable performance advantages.

**RankMatch** advances LDL with novel theory, strong empirical evidence, and clear practical benefits. We believe these characteristics meet NeurIPS standards for originality and impact, and we hope this perspective will assist the ACs in making their final decision.

---

### Decision · Program_Chairs · 2025-09-17

**Decision:**

Accept (poster)

**Comment:**

This paper introduces a novel contribution to label distribution learning. Initially, the reviewers raised concerns about the lack of experimental details. However, after the rebuttal, the authors successfully addressed most of these concerns: they investigated sensitivity issues, provided empirical guidelines for hyperparameter tuning for practitioners, implemented comparisons with additional SSLDL baselines, added missing loss ablations, elaborated on run-time costs, evaluated imbalance and applicability to large-scale datasets (e.g., Pascal VOC and COCO), and clarified notation issues as well as their method’s connection to related approaches. The AC has carefully reviewed the paper, the reviewer comments, and the rebuttal, and agrees that the paper is well-motivated, clearly written, and supported by thorough experiments. Therefore, the AC recommends acceptance. It is strongly encouraged that the authors incorporate all additional experiments and discussions from the rebuttal into the final version.